


ORCIDs:
Jianyang Xia (0000-0001-5923-6665)
**Title:**
TraceME (v1.0) - An online Traceability analysis system for Model Evaluation on
land carbon dynamics
**Authors:**
Jian Zhou[1], Jianyang Xia[1,*], Ning Wei[1], Yufu Liu[2,3], Chenyu Bian[1], Yuqi Bai[2,3], Yiqi
Luo[4]
**Affiliations:**
[1]Research Center for Global Change and Ecological Forecasting, School of Ecological
and Environmental Sciences, East China Normal University, Shanghai 200241, China
[2]Ministry of Education Key Laboratory for Earth System Modelling, Department of
Earth System Science, Tsinghua University, Beijing, 100084, China
[3]Joint Center for Global Change Studies (JCGCS), Beijing, 100875, China
[4]Center for ecosystem science and society, Northern Arizona University, Arizona,
Flagstaff, USA
[*]Corresponding author: Jianyang Xia (jyxia@des.ecnu.edu.cn)





**Abstract**
The synchronous increase of model complexity and data volume in Earth system
science challenges using observations to evaluate Earth system models (ESMs). The
challenge mainly stems from the untraceable of model outputs, the lack of automatic
algorithms, and the high computational costs. Here, we built up an online Traceability
analysis system for Model Evaluation (TraceME), which is traceable, automatic and
shareable. The TraceME (v1.0) can trace the structural uncertainty of simulated carbon
(C) storage in the state-of-the-art ESMs into gross primary production (GPP), carbon
use efficiency (CUE), baseline residence time and environmental scalars (temperature
and precipitation). The cloud-based framework used in TraceME provides the scientific
workflows and a shareable platform to achieve the automated analysis and distributed
data storage to greatly improve the efficiency of model evaluation. Then, we set up a
worker node in TraceME (v1.0) to store the data from Coupled Model Intercomparison
Project (CMIP6), and submitted tasks through browser to analyze the uncertainties of
CMIP6 models in the TraceME system. Overall, this new tool can greatly facilitate
model evaluation to identify sources of model uncertainty and provide some new
implications for the next generation of model evaluation.

## 1. Introduction

Inter-comparisons among Earth system models (ESMs) as well as between ESMs and data are an essential process to understand the performance of models, reduce their uncertainty, and provide a clear roadmap for model development (Todd-Brown et al., 2013; Eyring et al., 2016a; Getz et al., 2018). As both of the complexity of ESMs increases and the data volume expands rapidly in recent years, the ESMs' evaluation faces many new challenges. For example, the traditional methods used in model evaluation, mainly using statistical approaches, generally treat all metrics equally and ignore their indirect effects on model performance (Schwalm et al., 2010; Xia et al., 2013). Eyring *et al*. (2019a) has suggested that it is suboptimal to give each model equal weight in model evaluation because it is not independence among models. Moreover, model structure contributes approximately 80% of the variance in simulating the land carbon (C) cycle (Bonan and Doney, 2018; Bonan et al., 2019). The climate forcings and model parameters also contribute considerable uncertainty to the performance of ESMs (Ahlström et al., 2012; Shi et al., 2018; Luo and Schuur., 2020). These challenges call for new approaches of model evaluation which can systematically trace and quantify the structural sources of the uncertainty of the componentized models. In addition, the dramatically increase of data in observation and simulation pushes ecological research into a data-rich era (Luo et al., 2011), making it difficult for individuals to do research entirely locally to meet the computational requirements. Thus, an automated computation and shareable platform become essential for a rapid and comprehensive model evaluation. In general, the future approach of model evaluation requires many new characteristics, such as traceable, automatic and shareable.

A few efforts have been made to develop new analytical tools for evaluating ESMs, such as the International Land Model Benchmarking (ILAMB) System (Hoffman et al., 2016; Collier et al., 2018), the ESMValTool as a community diagnostic tool with performance metrics for evaluating ESMs (Eyring et al., 2016b), and the Land surface Verification Toolkit (LVT) (Kumar et al., 2012). These analytical tools mainly use many





statistical methods and multiple observations as benchmarks to evaluate the complex
ESMs. For example, the ILAMB system uses a set of statistical methods to construct a
scoring system based on observations as benchmarks to reflect the uncertainties among
ESMs (Collier et al., 2018). This benchmarking framework can directly demonstrate
the ability of models to simulate given ecological variables through its scores.
ESMValTool provides a very comprehensive model evaluation system for ESMs using
model outputs from the Coupled Model Intercomparison Project (CMIP) (Eyring et al.,
2016b). The LVT can fuse more information to evaluate land surface models, such as
remote sensing products and land information system (Kumar et al., 2012). These
model evaluation tools can effectively assess the differences between models and
observations, as well as the uncertainty among ESMs. Currently, these tools have not
yet focused on tracing the uncertainties in land models to their sources in model
structures, parameters and external forcings.
A traceable model evaluation tool is featured by its ability to systematically
quantify model uncertainty source. The traceability analysis method developed by Xia
*et al*. (2013) and Luo *et al*. (2017) is a systematic and effective approach to diagnose
the uncertainties of terrestrial C-cycle models. It decomposes the C dynamics into C
storage and C storage capacity, and uses C storage potential to represent the difference
between them. Then, those three variables can be further decomposed into a few
traceable components to trace the sources of model uncertainty, such as net primary
productivity (NPP), C residence time and environmental factors (temperature and
precipitation). This framework has been applied to some model evaluation studies
(Rafique et al., 2016; Jiang et al., 2017; Rafique et al., 2017). For example, Xia *et al*.
(2013) applied this framework to analyze the differences in modeled C processes among
biomes and the effect of nitrogen processes. Du *et al*. (2018) explored the effect of three
different carbon-nitrogen coupling schemes on C storage capacity and its responses to
atmospheric $CO_2$ enrichment. Zhou *et al*. (2018) applied the traceability analysis to
compare the simulated terrestrial C cycle across 25 models in three MIPs (i.e., CMIP,
TRENDY, and MsTMIP). Overall, this traceability analysis framework has the



advantage of providing a simple way to explain model variations by using a few
traceable components (Xia et al., 2013). Developing it as an available tool for model
evaluation can effectively trace and quantify the structural sources of uncertainty in
models.
Traditional model evaluations need to download large volumes of data from
multiple data centers to analyze it locally. For example, the individual users have to
repeatedly download model outputs of CMIP5 and CMIP6 from the servers of Earth
System Grid Federation (ESGF) for different analyses. However, the data volumes of
model outputs and data products both have been increased rapidly in the recent years.
For example, the size of database has been increased from 36 TB in CMIP3 to 2.5 PB
in CMIP5, and the volume of climate data is expected to 350 PB by 2030 (Overpeck et
al., 2011). Thus, it is more and more time-consuming for future researchers to download,
manage, preprocess and analyze the CMIP data on their local equipment (Xu et al.,
2019). To improve the computational efficiency of processing the data from distributed
data sources, it needs a new platform for model evaluation especially in the data
computing and storage. Bai *et al*. (2012) has shown that using "everything-shared-over-
the-web" to replace the common paradigm of "everything-locally-owned-and-operated"
is a promising solution to process distributed data. To achieve this goal, we need to
develop the model evaluation tools to be automatic and shareable platform. Thus, a
cloud-based framework with the scientific workflow is a good choice for model
evaluation. Cloud-based system can combine web-based technology to provide user-
friendly web interfaces and automatic workflows. Such web-based technology has been
used in the field of ecological modelling and model evaluation. For example,
Abramowitz. (2012) has introduced an online model evaluation tool, the Protocol for
Analysis of Land Surface models (PALS), to automatically evaluate the performance of
model. In addition, Huang et al. (2019) has developed a web-based software system
(i.e., Ecological Platform for Assimilating Data; EcoPad v1.0) to realize ecological
forecasting. The advantage of the web-based cloud technology can help the researchers
to focus on scientific problem of ESMs rather than processing the data.
The aim of this paper is to present an online traceability analysis system for model
evaluation (TraceME v1.0) to evaluate the ESMs based on the traceability analysis. We
first describe the technical aspects of the software system, include the traceability


method and data used in the tool, and then use part of the CMIP6 data as examples to
demonstrate the functionality of the TraceME. Finally, we discuss the implications of
TraceME (v1.0) for the next generation model evaluation and its future developments.

## 2. TraceME (v1.0):

### 2.1 Overview of the TraceME

TraceME (v1.0) is an online framework for automatically analyzing and evaluating the
performance of models using the traceability analysis method. It builds on a
collaborative analysis framework for distributed gridded environmental data (CAFE;
Xu et al. 2019), which consists of at least one central server and more than one worker
node. The central node is used to manage the descriptive information about each node,
and the data and the available analytic scripts are stored on each worker node. Each
node (center and work node) consists of web-based User Interface (UI), data index
module, task-managing module and data analysis module. This multi-node structure
can realize collaborative analysis of distributed data (More details are described in Xu
et al., 2019). TraceME inherits CAFE's ability to collaborate on distributed data, but
has different core functions and focuses (Fig. 1). It integrates the traceability analysis
and focuses on analyzing and tracing the sources of model uncertainty rather than the
flexible data preprocessing in CAFE. In addition, TraceME makes several technical
updates to accommodate the processing of multivariate data for the systematic analysis
of uncertainty of models. When a user selects the data of interest and sends a request
through the web browser, the scientific workflow is triggered. The corresponding tasks
are assigned by the central node to the worker node containing the corresponding data,
and then running the traceability analysis and returning the results to the user interface
(Fig. 1). The major components of Web-based UI, data analysis module and data
management module are described below.

The web-based UI provides a straightforward way for users to interact with the

system through a web browser. It can select data of interest, submit tasks, check the
status of tasks and present the results of traceability analysis. The registered users can
filter the data of interest by institute, model, frequency and other information of the
dataset. After submitting the task, the web-based UI sends requests to the connected
node and run the data analysis module. The results of traceability analysis will be saved





and a relational database is used to store that information. User can retrieve and
visualize the results of both figures and NetCDF files according to traceability analysis
through the web-based UI.
The data analysis module is to realize the traceability analysis, which can
systematically analyze the uncertainty of models and output the corresponding analysis
results. It consists of an analysis launcher, a command executor and the traceability
analytic script. When the real-time monitoring of the analysis launcher picks up the task,
it parses the information of task and instantiates it as a Java command executor. The
command executor invokes the analytic script written by Python to run the traceability
analysis.
The data managing module includes data index submodule and task managing
submodule. The data index submodule manages the descriptive information about data
(data file name, storage path and data attributes) stored on each worker node. Task
managing module is used to task submission, task dispatching, and task status/results
query services on each node. The data managing module in the central node is used to
maintain the global data and task information. User can scan and update data
information by the web-based UI supported by data index module. When user sends the
task-by-task managing submodule, the task information will be dispatched to a node
and maintained in the database on that node. The task managing submodule in the
central server provides global task information retrieval.

**2.2 Traceability analysis framework**

The core functionality of TraceME is based on traceability analysis framework of C
storage (X) at steady state that developed by Xia *et al*. (2013). This framework is
extended to transient dynamic by decomposing the C storage dynamics into a three-
dimensional parameter space (Luo et al., 2017). The latter can be further partitioned
into traceable components to track the sources of model uncertainty. In the framework
of Traceability analysis, terrestrial C storage is at dynamic disequilibrium, which is
collectively influenced by internal C-related processes, environmental forces, and their
interactions (Luo and Weng, 2011). Under given environmental conditions, the C
storage of an ecosystem can reach the steady state, which can be defined as C storage
capacity ($X_C$). In ESMs, we can obtain the $X_C$ by spinning up the model to the steady



state (Xia et al., 2012). Because the externally forces, such as climate, are never at
steady state, so the $X_C$ is always deviate from the realistic C storage in natural
ecosystems. Such deviation or difference between the transient C storage and $X_C$ was
defined as C storage potential ($X_P$) (Luo et al., 2017). Hence, the transient C storage of
an ecosystem can be determined by $X_C$ and $X_P$. Then, $X_C$ is jointly determined by
ecosystem C input (e.g., net primary production, NPP) and ecosystem C residence time;
($\tau_E$). As the net ecosystem C input, NPP is determined by gross primary production
(GPP) and C use efficiency (CUE). CUE describes the capacity of an ecosystem to
effectively absorb C from the atmosphere (DeLucia et al., 2007; Xia et al., 2017). The
$\tau_E$ can be further traced to the baseline C residence time ($\tau_E'$) and the environmental
scalar ($\xi$). $\tau_E'$ represents the ecosystem C residence time under optimal environmental
conditions, which is usually determined by the preset soil properties and vegetation
characteristic in the model (Xia et al., 2013). The $\xi$ is influenced by several factors,
such as climate, oxygen, and land cover. The climate is the most common limiting factor
in ESMs. In this study, we focus on the effect of climate forcing (i.e., temperature and
precipitation) on the $\tau_E'$. The detail of Traceability analysis method is described in Xia
*et al*. (2013), Luo *et al*. (2017) and Zhou *et al*. (2018).
In the framework of traceability analysis, land C storage is ultimately attributed to
its traceable components, which are related to the natural properties expressed by the
model (Fig. 2). For example, GPP is the photosynthetic property of vegetation; baseline
residence time is related to the soil attributes (Fig. 2). In order to quantify the
contributions of these traceable components to the uncertainty of models, we use a
hierarchical partitioning method (Chevan and Sutherland, 1991) to decompose the
uncertainty of simulated C storage dynamics. This method can be used to calculate the
independent effect of each explanatory variable ($x_1$, $x_2$, $x_3$ … $x_k$) on a single dependent
variable ($y$). The independent effect of $x_l$ ($I_{xl}$) means the contribution of $x_l$ to the variable
$y$, which is calculated by comparing the fit of all models ($2^k$ possible models) including
$x_l$ to that lacking $x_l$ by the hierarchical partitioning (Chevan and Sutherland, 1991;
Murray and Conner, 2009). In our system, we calculate the variance contribution of the
variables using the 'hier.part' package in R. First, the C storage can be decomposed into
carbon storage capacity and potential. The relative contribution of $X_C$ and $X_P$ to X are
estimated. Second, the carbon storage capacity is decomposed into NPP and residence




time. To apply this method, all variables are their logarithmic form: $\ln(X_C)$, $\ln(NPP)$
and $\ln(\tau_E)$. The contributions of NPP and $\tau_E$ to $X_C$ are calculated. Third, NPP is
further decomposed into GPP and CUE, and residence time is decomposed into baseline
residence time and environmental scalars (temperature and precipitation). Convert them
into logarithmic form. The contributions of GPP and CUE to NPP are calculated. The
contributions of baseline residence time, temperature and precipitation to residence
time are calculated as the same way. Finally, the contributions of these traceable
components (GPP, CUE, baseline residence time, temperature and precipitation) can be
calculated.
**2.3 Data**
In this study, the TraceME (v1.0) used CMIP6 model outputs as examples to describe
the workflow of this platform. The TraceME can be compatible with any model output
that follows the Network Common Data Format (netCDF) Climate and Forecast (CF)
Metadata Convention (http://cfconventions.org/). The data is stored in the database of
each node, and the information of data in each node is aggregated to the central node,
where users can access and handle all data stored on all nodes of the whole system. On
the other hand, TraceME (v1.0) is a systematic framework for uncertainty analysis on
the terrestrial carbon cycle for CMIPs. It requires a multivariable dataset to analyze and
trace the sources of uncertainty in simulating ecosystem carbon storage. The time series
data of total ecosystem carbon storage are needed, which generally consist of vegetation
carbon (leaf, woody and root carbon pools), soil carbon (fast, slow and passive soil
carbon pools) and/or litter carbon pools (litter and/or coarse woody debris) in the model
outputs. The time series data of NPP, GPP and forcing data (temperature and
precipitation) are also used for further model intercomparisons. All data used in this
study is from 7 CMIP6 models (the release data before July, 2019) and collected from
ESGF (http://esgf.llnl.gov/) as shown in Table 1.
**3. Applications of TraceME (v1.0)**
**3.1 Temporal dynamics of land carbon storage in CMIP6 models**
TraceME (v1.0) provided an automatic traceability analysis for data of temporal interest,
which can be used to evaluate the temporal dynamics of land C storage simulated by



models. We used 7 models that had been submitted results in CMIP6 to analyze the
uncertainty of these models in simulating historical land carbon storage from 1850 to
2014. Once we selected the data of interest through the browser and submitted the task,
the daemon automatically preprocessed the data and ran the temporal traceability script,
and returned the results in the forms of figures and data in netCDF format. Under the
traceability analysis system, the temporal dynamics of global annual C storage
simulated by different models were first calculated (Fig. 3a). The global annual C
storage varied greatly among the 7 models, ranging from 938.76±11.36 to
2206.76±50.14 Pg C (Fig. 3a). Decomposing the C storage into C storage capacity and
potential, the C storage potential ranged considerably from about -21.66±54.39 to
58.07±57.62 (Fig. 3a). And the C storage capacity of different models in response to
external force was also quite different. For example, the lowest simulated C storage
capacity was IPSL-CM6A-LR during 1850 to 2014, which was 944±27.14 Pg C, and
the other models were from about 1677.57±57.21 to 2263.43±106.61 Pg C (Fig. 3a). To
further analyze the uncertainty of C storage capacity, this framework decomposed it
into NPP and residence time. These two variables reflected the net C input capacity
(38.48±2.72 to 68.74±5.88 Pg C yr$^{-1}$) and the C turnover time of ecosystem (23.22±1.75
to 56.23±3.10 years) in the models (Fig. 3b-c and 4a). In details, the lowest simulated
NPP was CESM2 and the shortest residence time was IPSL-CM6A-LR, while
CanESM5 had the largest NPP and residence time among all models (Fig. 3b-c and 4a).

To further trace the uncertainty sources of NPP simulated by models, TraceME

(v1.0) decomposed it into GPP and CUE (Fig. 3d-e and 4b). The differences of GPP
and CUE in different models reflected the model's photosynthetic capacity and C
transfer efficiency from atmosphere to ecosystem biomass. Based on this process,
TraceME could quantify the effects of models simulating photosynthesis and
respiration on the uncertainty of NPP. For example, NPP simulated by CanESM5 and
EC-Earth3-Veg had larger uncertainty, which were 68.74±5.88 and 48.96±2.78 Pg C yr$^{-}$
$^{1}$ respectively during 1850 to 2014, whereas their GPP was similar, which were
132.22±8.18 and 127.72±4.38 Pg C yr$^{-1}$ respectively (Fig. 3b-e and 4b). Therefore, the
uncertainty of NPP between the two models mainly came from CUE (0.52±0.01 and
0.38±0.02, respectively), which was related to autotrophic respiration. In addition,
residence time was traced to baseline residence time and environmental scalars in
TraceME. Baseline residence time explained the uncertainty of some preset attributes
in the model structure, such as soil C decomposition rate, and the environmental scalar
reflected the impact of external forces on the performance of model. For example,
IPSL-CM6A-LR had the shortest residence time (23.22 1.75 years) than other models
during 1850 to 2014, and compared with external forces, the main reason was it had the
shortest baseline residence time (18 years) among all models (Fig. 3c, 3f-i and 4c).
Hence, the development of IPSL-CM6A-LR was suggested to pay more attention to
some preset attributes of soil. Furthermore, the environmental scalar in TraceME here
was the global annual scale. Its uncertainty reflected the variability of interannual
variation of temperature and precipitation used in each model over all models rather
than the direct difference of external forces among models (Fig. 3f-h and 4c-d).
Overall, after analyzing the uncertainties of all traceable components, TraceME
summarized the variance contributions of the components to the uncertainty of land C
storage among models. This framework traced the uncertainty of land C storage to
several sources, and the hierarchical partitioning method could be used to decompose
the variation in it into the traceable components. For example, the variation of land C
storage among 7 CMIP6 models was mainly from residence time and NPP, and the C
storage potential contributed about 4.5% (Fig. 5). Comparing all traceable components,
the variation in C storage simulated by these models was dominated by baseline
residence time (Fig. 5).
**3.2 Spatial distribution of land carbon storage uncertainties in CMIP6 models**
TraceME (v1.0) provided the ability to analyze spatial uncertainty of models. It could
trace the sources of the uncertainty of models in simulating C storage at each grid. The
region of interest in TraceME (v1.0) could be selected by latitude and longitude. Here,
we selected global data of 7 CMIP6 models by setting the spatial range according to
longitude and latitude through the browser and submitted this task of spatial traceability
analysis. When the task was submitted, TraceME (v1.0) extracted data from the entire
system for processing and called for spatial traceability analysis scripts. The mean
spatial pattern of the 7 models showed C storage in boreal regions was higher than in
other regions (Fig. 6a). However, some models, such as IPSL-CM6A-LR, had no such
spatial pattern (Fig. 7), and the high variability of C storage simulated by these models
also appeared in the boreal regions, such as Siberia and northern North America (Fig.
6b). To further research the sources of the uncertainty of models in simulating C storage,


TraceME (v1.0) provided the spatial patterns of C storage capacity and C storage
potential (Fig. 6c-f and 7).

According to traceability framework, TraceME (v1.0) provided the spatial

distributions of NPP and residence time to explain the uncertainty of land C storage
capacity among models (Fig. 7). From the results of 7 CMIP7 models, the distribution
of the variation in NPP among these models occurred in the lower latitude region, while
the variation of residence time was mainly distributed in northern high latitude region
(Fig. 8a and 8d). Following the workflow of TraceME (v1.0), the uncertainties of global
distributions of NPP and residence time were further decomposed into the spatial
variations of their traceable components: GPP, CUE, baseline residence time and
environmental scalars (Fig. 8b-c and 8e-f). To better guide model development, it is
important for model evaluation to provide the information of the spatial distribution of
the dominant factor influencing the simulation of land C storage. TraceME (v1.0) could
analyze the variation contributions of all traceable components to land C storage at each
grid, and offered the spatial pattern of the dominant factor (Fig 9). For example, the
baseline residence time and GPP were the major contributors to the global distribution
of the variation of simulated C storage by the 7 models from CMIP6 (Fig. 9). Compared
to GPP, baseline residence time dominated the uncertainties of simulated land carbon
storage in northern high latitude, eastern Asian and the northern part of South America
(Fig. 9).
**3.3 Uncertainty analysis of simulated carbon storage from models at different**
**periods**
Assessing the performances of model over different periods could provide a more
comprehensive understanding of the model's ability to simulate land C storage. For
example, the environmental scalars among the 7 CMIP6 models had larger variability
at initial state (e.g. from 1850 to 1860) than those at current state (e.g. 2004 to 2014)
(Fig. 3f). It was necessary to research in detail the sources of uncertainty that different
models simulated at different periods. It was convenient for TraceME (v1.0) to submit
multiple tasks and perform them simultaneously. We submitted four tasks for temporal
and spatial analysis of the performance of 7 CMIP6 models at two periods (1850 to
1860 and 2004 to 2014 presenting initial and current conditions respectively). From the
results, the dominant contributor of initial state of models was baseline residence time





that was similar to that at current period (Fig. 10). The variance contribution of C
storage potential to C storage simulated by the models at the two periods had larger
difference, which was 5.2% and 19.1% at initial and current periods respectively (Fig.
10). In addition, GPP and residence time were also the major contributors to the global
distribution of the uncertainty of simulated land C storage at the two periods (Fig. 10).
However, the regions where GPP was the dominant contributor of carbon storage
variability at initial period were larger than that at current period, especially in the high
northern latitudes (Fig. 10).
**4. Discussion**
**4.1 Facilitating the next generation of model evaluation**
The increase of model complexity and the expansion of observation promote the model
evaluation into the next generation. In our study, we propose that the next generation of
model evaluation needs to some new characteristics, including traceable, automatic and
shareable. TraceME (v1.0) is designed to meet these three characteristics, and can
provide complementary functions to those existing model-evaluation tools. For
example, ESMValTool (v1.0) uses observational data (e.g. observations for Model
Intercomparison Projections and re-analyses data, obs4MIPs and ana4MIPs) as
diagnostics and performance metrics to measure the uncertainty in ESMs (Eyring et al.,
2016b). ILAMB constructs a comprehensive set of observation data (e.g. Fluxnet and
MODIS) as benchmarks and a scoring system to evaluate the performance of land
models (Collier et al., 2018). As the core function of TraceME, the traceability analysis
is helpful for extending current model evaluations to quantify the structural sources of
the uncertainty of model (Lovenduski et al., 2016). Rather than simply comparing the
differences in simulated C storage among models, this method can trace the
uncertainties to the carbon storage potential, GPP, CUE, baseline residence time and
environmental factors (temperature and precipitation), and quantify the relative
variance contributions of these traceable components (Fig. 4 and 8). For example, the
annual C storage simulated by IPSL-CM6A-LR is much lower than other models, and
TraceME can track it to C storage capacity (Fig. 3a). After a further systematic analysis
on C storage capacity, TraceME tracks the low estimates on the global scale in IPSL-
CM6A-LR to C residence time, especially the baseline C residence time (Fig. 3-4).
Thus, TraceME can not only show the structure sources of the disagreement on global





C storage between ESMs, but also identify the key uncertain component for a specific
model to facilitate its development.
The cloud-based framework adopted by TraceME (v1.0) provides a web-based
scientific workflow and shareable platform for automated computation. Compared with
the rapid acquisition of observational data, the slow development of ESMs has become
one of the bottlenecks to a deeper understanding of ecosystem. As an important part of
model development, model evaluation also needs higher computational efficiency. In
the absence of automated computation, model evaluation is usually computationally
low-efficient due to the repeated computation for each model output. Therefore,
automation is a crucial property for an efficient model evaluation. Most model
evaluation tools have implemented automation by encapsulating workflows as offline
software packages. For example, both ILAMB and ESMValTool have released their
second version packages (Collier et al., 2016; Eyring et al., 2019b). TraceME (v1.0)
uses the web-based technology to integrate a user-friendly interface and automated
computation in background. Users can complete all steps of data processing including
submitting task, processing data and managing results through a web browser with a
unique ID and web address. The web-based workflow has the advantages of
convenience, timeliness and visualization (LeBauer et al., 2013), avoiding the need for
technical training for scientific researchers to run packages.
Both modeling outputs and observation data come from multiple data sources. For
example, model comparison projects have data sources of CMIP, TRENDY and
MISMIP. As shown by Song et al. (2019), more than one thousand global-change
experiments have been done in the ecology field to monitor the responses of terrestrial
C processes to global change. In order to more fully evaluate the performance of models,
researchers need to collect large amounts of data from different data sources. The cloud-
based technology is considered to be the most effective means to solve the distributed
geospatial big data (Bai and Di, 2012; Li et al., 2016). TraceME (v1.0) uses the cloud-
based framework that consists of a center node and multiple worker nodes set at
different data sources, and the user can use and share the data in this system. With the
increase in the amount of model simulations and observations, and the tediousness of
processing data, the shareable approach would be a good way to improve the efficiency
of model evaluation. Meanwhile, it can help researchers who develop models focus



more on the scientific issues rather than the technical problems.

**4.2 Future work**

Although TraceME (v1.0) provides a complete and comprehensive system for model
evaluation, there are still several aspects must be developed and this work is ongoing.
The first one is the traceability analysis method used in TraceME (v1.0). In our current
version of TraceME, NPP is finally decomposed into GPP and CUE. However, Xia et
al. (2015) has shown GPP is joint controlled by plant phenology and physiology, and it
can be decomposed into the carbon dioxide uptake period (CUP; number of days per
year) and the maximal daily rate of gross photosynthesis during the CUP ($GPP_{max}$) that
represents a property of plant canopy physiology. $GPP_{max}$ is a critical indicator to
quantify the capacity of terrestrial ecosystem productivity (Huang et al., 2018). CUP is
related to phenology, which is mainly influenced by environmental factors, such as
temperature and water availability (Jaworski and Hilszczański, 2013; Xie et al., 2015;
Piao et al., 2019). In addition, Cui et al. (2019) indicates that GPP can be further
explained by the subsequent carbon cycle processes and related vegetation functional
properties, such as leaf area index and leaf-level photosynthesis. Other environmental
factors also affect carbon residence time and NPP, such as atmospheric $CO_2$, land-use
change, and nitrogen availability (Tian et al., 1999; Wu et al., 2003; Melillo et al., 2011;
Van Groenigen et al., 2014; Wieder et al., 2015). These traceable processes can be
further added to the traceability analysis framework and applied to TraceME.
Secondly, the current version of TraceME focuses on the comparative analysis
among multiple models and does not use observation data as benchmarks to analyze
model uncertainty. Since the traceability analysis is a systematic analysis method, it
requires the time-series observations of all variables used in this system to form a
complete benchmarking dataset, such as NPP, GPP and/or net ecosystem exchange
(NEE). Some model evaluation systems (e.g. ILAMB and ESMValTool) have built
large datasets of observation data (Eyring et al., 2016b; Collier et al., 2018). Particularly,
in TraceME, residence time is an important variable for the traceability analysis, and
more efforts are still needed to construct a global database of measured C residence
time. Wang et al. (2019) have constructed a global soil C residence time database, and
used it to evaluate the simulated mean soil C transit times by ESMs. More works are
needed to develop the database for TraceME. On the other hand, observed data may





have different spatial scales ranging from globe to site, so the future version of TraceME
should adapt model evaluation at different scales. Some recent studies have applied the
traceability method to analyze the land C storage dynamic at different scales. For
example, Jiang et al. (2017) has applied the transient traceability analysis method to
compare the difference in ecosystem C dynamics between Duke forest and Harvard
forest. Cui et al. (2019) has analyzed the performances of MsTMIP models in
simulating ecosystem productivity in the East Asian monsoon region. These analyses
could be efficiently applied with the TraceME if the datasets are implemented in the
future versions.

Lastly, the cyberinfrastructure of TraceME (v1.0) is derived from CAFE. CAFE is

a multi-node collaborative platform that can increase the efficiency of performing batch
analyses and comparing data from multi-node (Xu et al., 2019). To install CAFE
software package in more data centrals is an important goal of the development of
CAFE, and it involved many computer techniques. For example, Java, Tomcat and
MySQL running in a Linux environment are necessary for a CAFE node, and some
tools, such as NetCDF Operators (NCO) and Climate Data Operators (CDO), are
expected to fulfill data analysis (Xu et al., 2019). Moreover, to better accommodate
more data centers, some aspects of CAFE also need further improvement and
development. For example, the community tools for publishing new analysis functions,
version-control mechanism, intermediate analysis result, and encryption techniques
(Xu et al., 2019). The infrastructure of TraceME inherits from CAFE and it is expected
to evolve into a more open community for users and developers. These problems in
CAFE also need to be addressed in TraceME. Developing more worker nodes is also
the inherent requirements for the shareable trait of TraceME, and we also need to
develop the infrastructure of TraceME to adapt more data centers. For example, CAFE
cannot directly process data from multiple databases on different nodes in a single task
because it does not currently have this requirement (Xu et al., 2019). However, in the
system of TraceME, there is a need to compare models across data sources, such as
models between TRENDY and CMIP. We are working to develop TraceME to support
for accessing multiple databases from different nodes in one task. One possible solution
is to develop standard interfaces for the results of traceability analysis method on each
node, and then aggregate them into one node for the final comparative analysis to
reduce data transfers between different nodes. Moreover, the databases in TraceME



(v1.0) need to be updated in a timely and automated manner, especially the amount of
benchmarking data products is increasing rapidly (Hoffman et al., 2016). Updating
databases more convenient is also a requirement for TraceME's automated computing.
Overall, we hope that TraceME can provide a new tool to evaluate global land models
and drives the model evaluations on terrestrial biogeochemistry towards traceable in
the near future.

**Code availability**

The code for the traceability analysis is uploaded on
https://doi.org/10.5281/zenodo.3766626.

**Author contributions**

JX and JZ designed this study. JZ build the system of TraceME (v1.0). NW
provided the support of some algorithms in the system. YB and YF provided the code
and technical support of CAFE. JZ wrote the first draft, and all other authors contributed
to revision and discussion of the results.

**Competing interests**

The authors declare that they have no conflict of interest.

**Acknowledgements**

This work was financially supported by the National Key R&D Program of China
(2017YFA0604600) and National Natural Science Foundation of China (31722009).
We acknowledge the World Climate Research Program (WCRP) that is responsible for
CMIP, and we thank the modelling groups for providing their model output.

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





**Table 1** The list of seven ESMs used in this study from CMIP6.

| ESM | Land Model | Variables |
|---|---|---|
| BCC-ESM1 | BCC-AVIM2 | |
| CanESM5 | CLASS-CTEM | GPP, NPP |
| CESM2 | CLM5.0 | Total vegetation C pool (cVeg) |
| IPSL-CM6A-LR | ORCHIDEE | Total litter C pool (cLitter) Total soil C pool (cSoil) |
| MIROC-ES2L | VISIT-e | Precipitation (pr) |
| CNRM-ESM2-1 | ISBA | Temperature (tas) |
| EC-Earth3-Veg | LPJ-GUESS | |




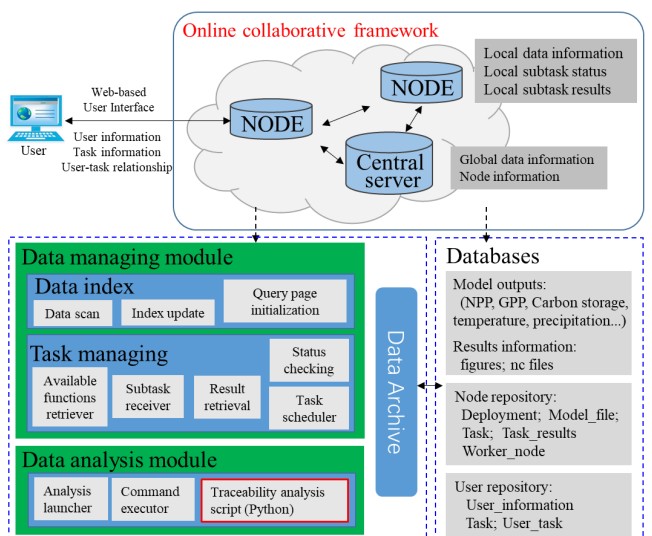


Figure. 1 Schematic overview of TraceME (v1.0).



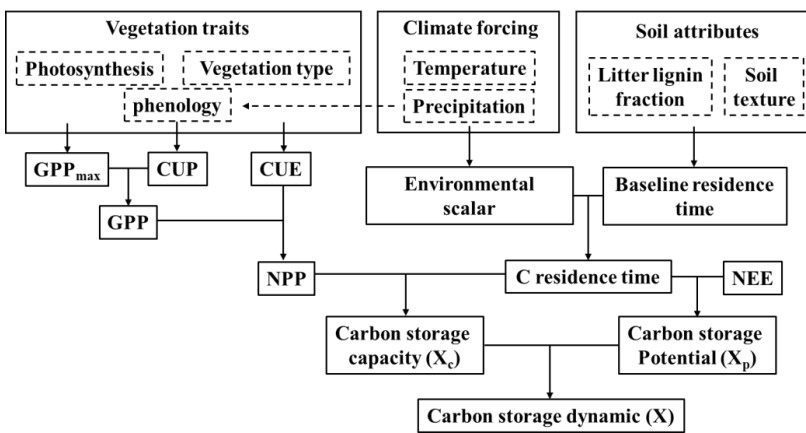


Figure. 2 The theoretical framework of traceability analysis. The transient carbon
storage dynamic can be decomposed into carbon storage capacity and potential. Then
the NPP and residence time can explain the carbon storage capacity. NPP can be traced
to GPP and carbon use efficiency (CUE). Residence time can be traced to environmental
scalars and baseline residence time. These traceable components can be explained by
related attributions.



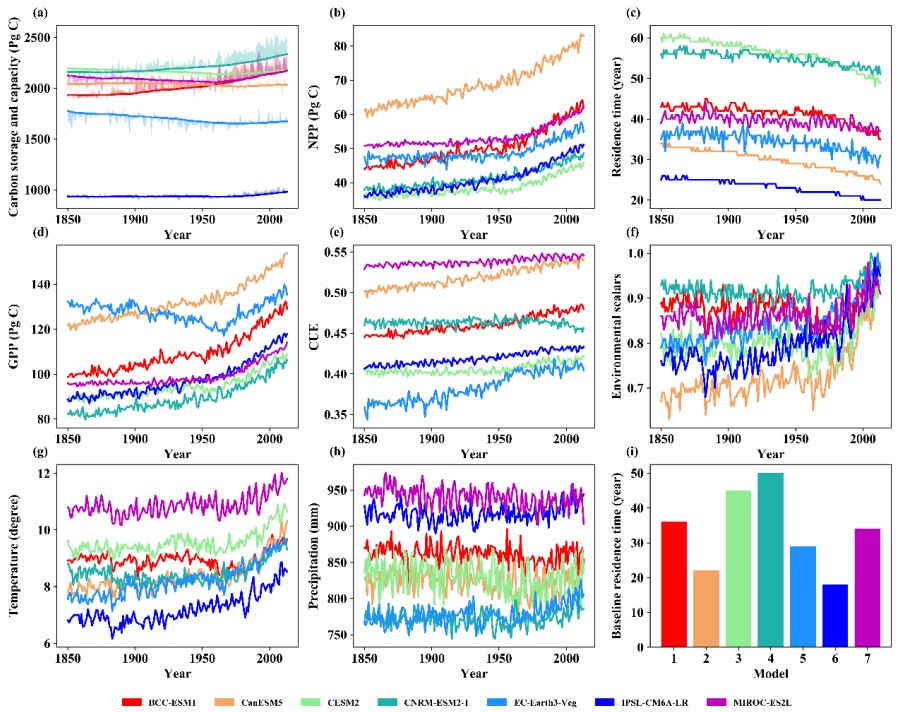

Figure. 3 The time series of annual carbon storage (solid lines) and carbon storage capacity (the contour lines) (a), and the traceable components: (b)-(h) for NPP, residence time, GPP, CUE, environmental scalars, temperature and precipitation simulated by 7 CMIP6 models, respectively. (i) is the baseline residence time for each model. The shades in (a) represent the annual variation in carbon storage potential for models (positive above the soil lines, and negative below the solid lines).



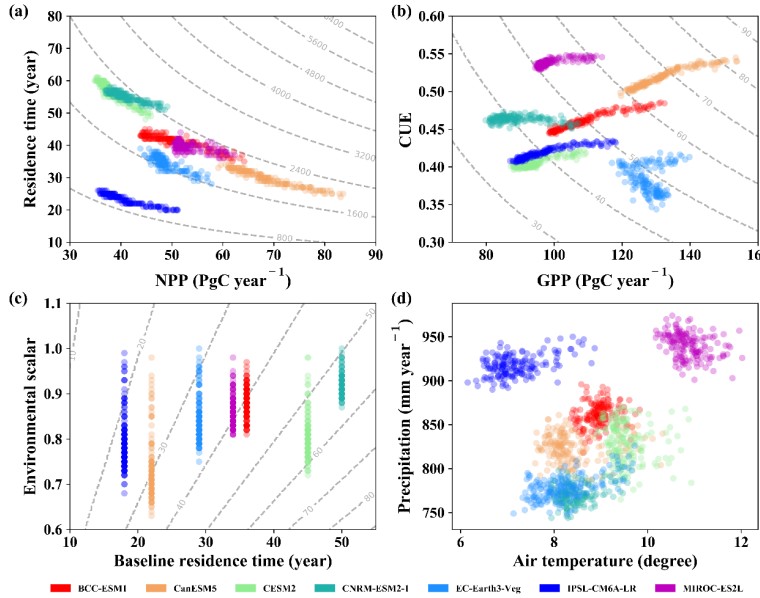


Figure. 4 The traceability decomposition of carbon storage capacity. The contours lines

in (a)-(c) represent carbon storage capacity, NPP and residence time respectively. Points

represent the global annual values for variables.




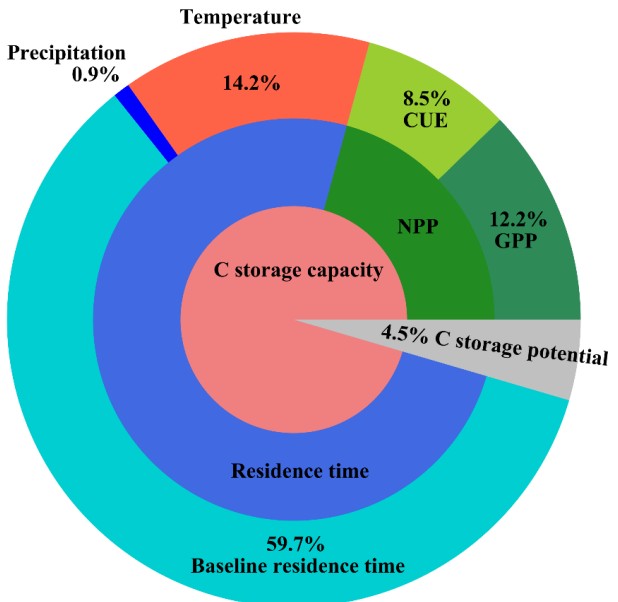


Figure. 5 Variation decomposition of the carbon storage based on annual data from
models (CMIP6). The inner circle indicates the carbon storage is composed into carbon
storage capacity and carbon storage potential, and their variance contributions. The
middle circle represents the carbon storage capacity is decomposed into NPP and
residence time, and their variance contributions. The outside circle indicates that the
NPP is decomposed into GPP and CUE, and residence time is decomposed into baseline
residence time and environmental scalars (temperature and precipitation), and their
variation contributions to carbon storage.



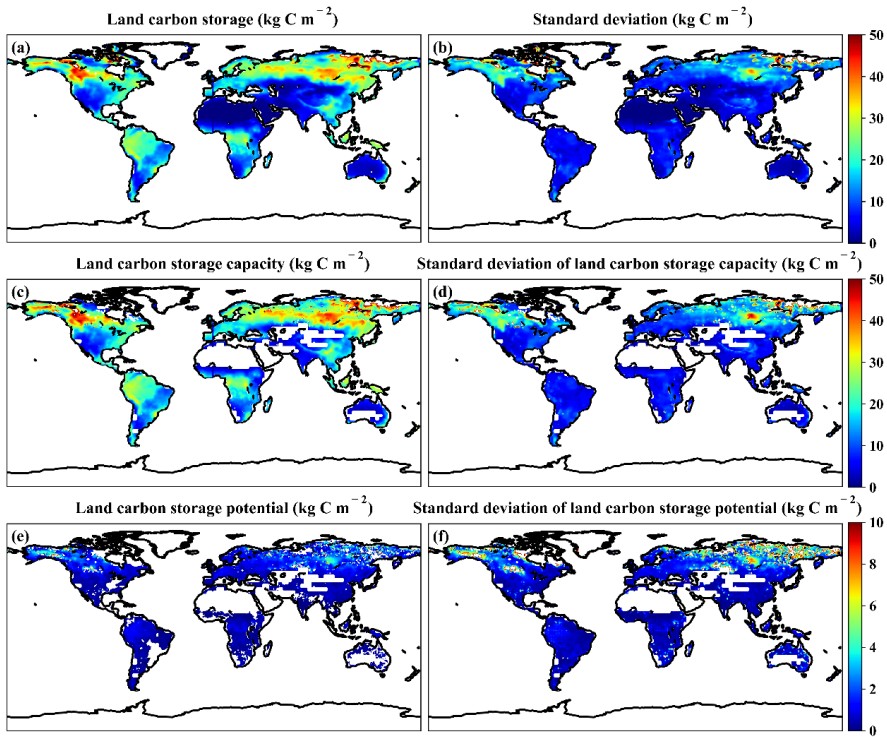

Figure. 6 The spatial distribution of the mean land carbon storage (a), land carbon storage capacity (c) and potential (e) simulated by 7 models from CMIP6 during 1850 to 2014, and the standard deviation of land carbon storage (b), land carbon storage capacity (d) and potential (f) from these models.

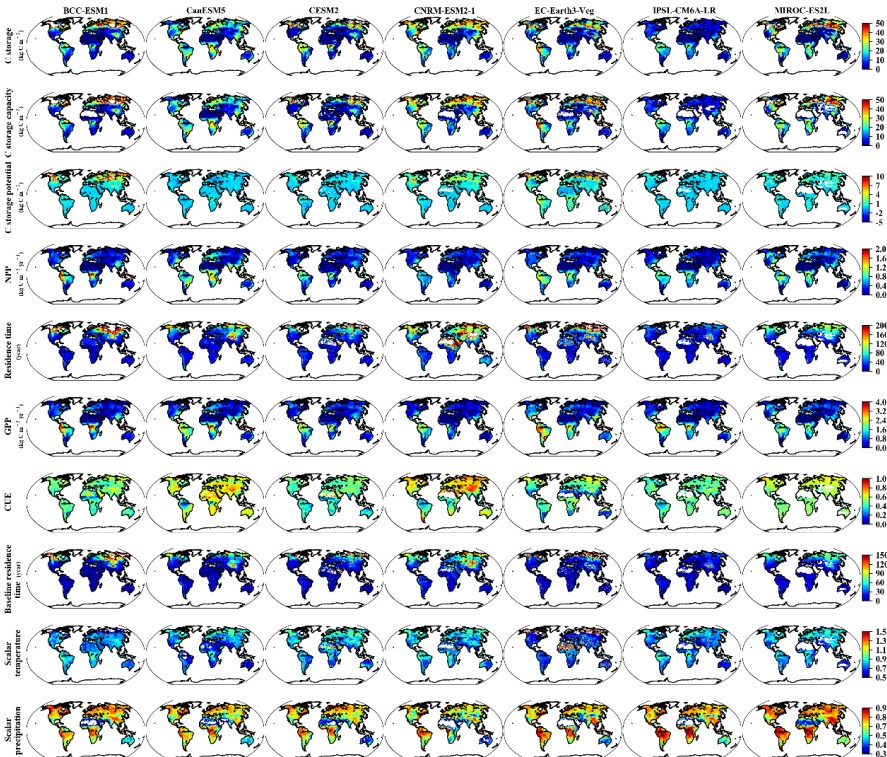


Figure. 7 The mean of carbon storage and its traceable components: carbon storage
capacity, carbon storage potential, NPP, residence time, GPP, CUE, baseline residence
time and scalars (temperature and precipitation) simulated by 7 CMIP6 models for the
historical period 1850-2014.





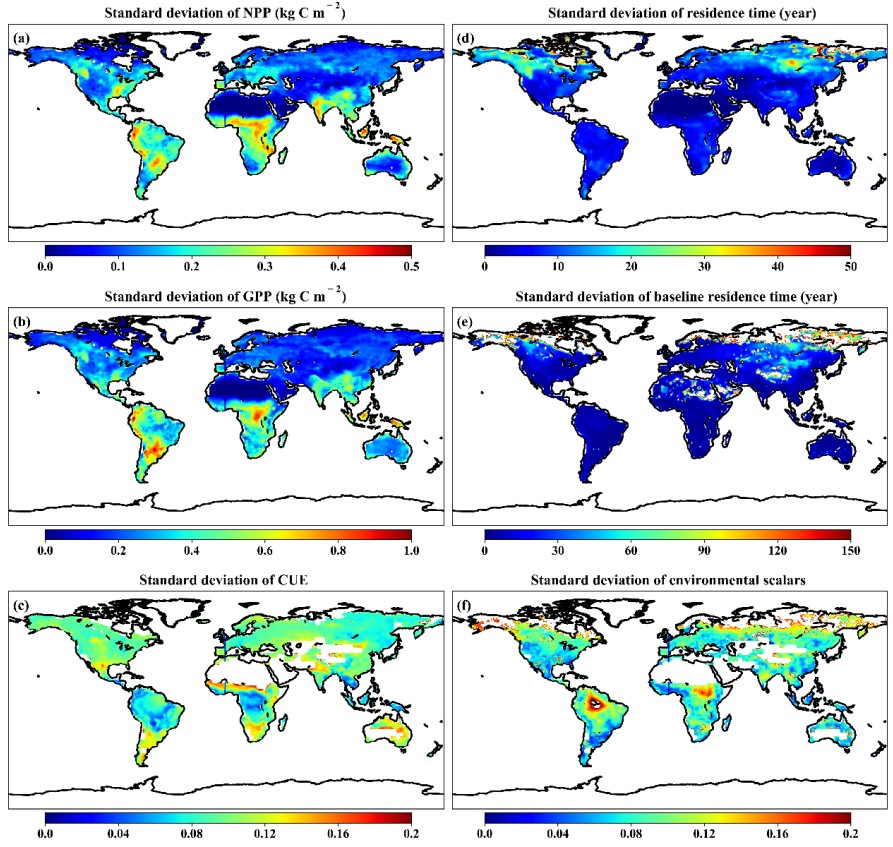


Figure. 8 The global distribution of the variations of the traceable variables simulated by 7 models from CMIP6 for the historical period 1850-2014. (a)-(f) represent the standard deviation of NPP, GPP, CUE, residence time, baseline residence time and environmental scalars, respectively.


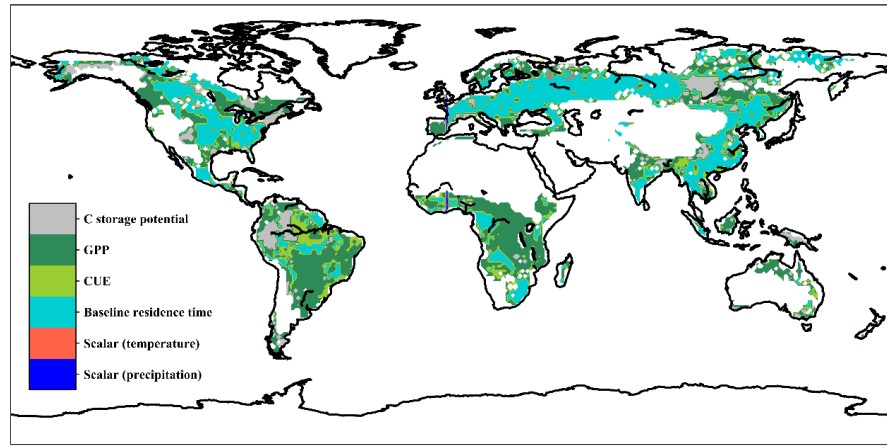


Figure. 9 The global distribution of the dominant variable for the variation in simulated
land carbon storage by the models from CMIP6 during 1850 to 2014.

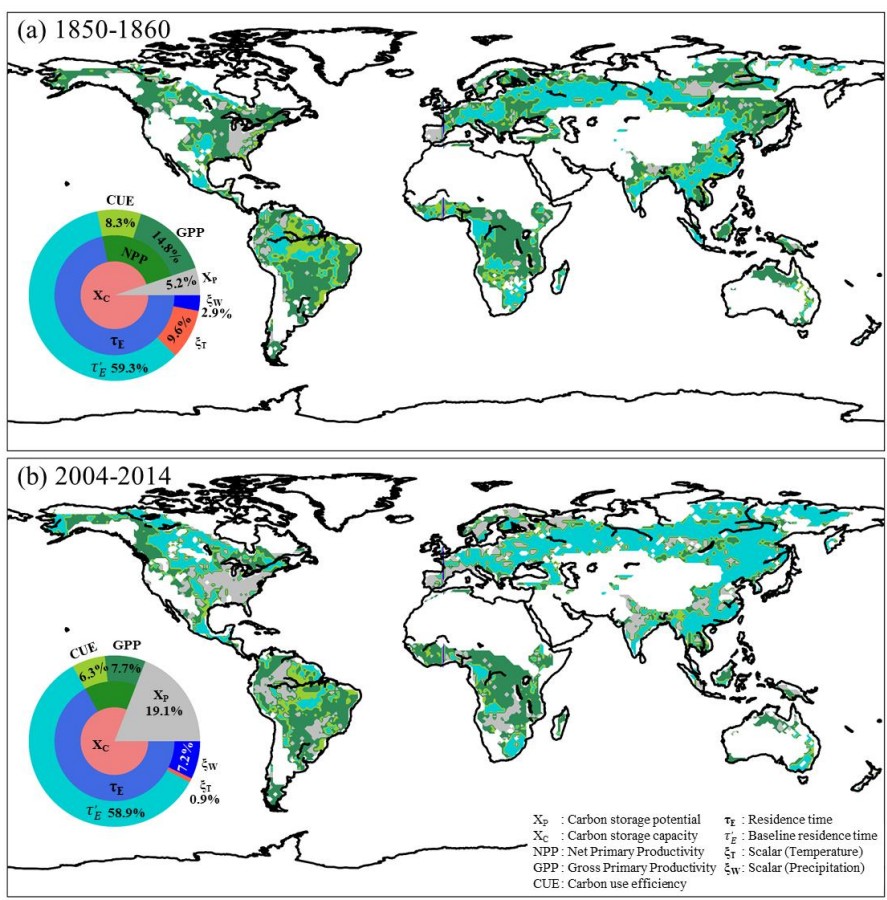

Figure. 10 The traceability analysis results of land carbon storage simulated by 7 models from CMIP6 at different periods: (a) 1850-1860; (b) 2004-2014. The subplot of each panel is the variation decomposition of the carbon storage based on annual data.