# Peer review of "Jianyang Xia (0000-0001-5923-6665)"

_Geoscientific Model Development, 2020_

## Referee Comment (RC1) · Anonymous Referee #1 · 15 Jun 2020

General Comments

This paper discusses the so-called traceability analysis that the authors have published and applied since 2013. The motivation for the paper is the development of a cloud-based software built on the CAFE framework which implements their method and is launched via web interface. The authors position their analysis as the answer to what they consider are the deficiencies in the current state of model evaluation, describe how their software and algorithms work, and then show analysis results from a selection of CMIP6 models.

I have two reservations against this paper. The first is that the novel portion of the pa-

per, the TraceME software, is not a substantial advance in modeling science. TraceME is a web interface to an analysis script which runs on their server. There is some sophistication in that source data can exist on multiple nodes, but this is the CAFE framework and not TraceME itself. The authors neither provide a link to test out the operability of the software they describe or even a screenshot of the user interface. The reader is left to trust the authors that this software exists and is functional as they say it is.

My second reservation is that a significant portion of the paper puts forth a viewpoint of the state and needs of model evaluation that is poorly supported. The authors use ambiguous terms such as 'traceable', 'shareable', 'indirect effects','high computational cost', and 'automatic'. They use these terms to describe their viewpoint on the deficiencies of current model evaluation, but do not say in detail what these terms mean.

In what follows I will specifically refer to lines which have led me to this opinion of their work.

Specific Comments

lines 22-24: You assert that the main challenge of using observations to evaluate ESMs is the untraceability of model outputs. It is not clear to me what this means precisely or why it is true. Why is it the 'main' challenge among many others?

lines 44-45: What does 'generally treat all metrics equally' mean? The referenced IL-AMB package, for example, does not treat every statistical measure equally. Neither does it treat measures from all data sources equally. Also, what do you mean by 'indirect effects'? ILAMB also considers variable-to-variable relationships including metrics such as Koven's inferred carbon turnover time [1]. I am having trouble envisioning what the authors mean by this statement.

lines 56-59: It does not follow that 'an automated computation and shareable platform' is essential because of a increase in the amount of data. Environmental computing

has utilized computing centers for many decades in response to computational and data costs. Furthermore, these computing centers are becoming more user-friendly. For example, NERSC now supports Jupyter notebooks [2] which allow you to script analysis on your browser without needing to move data around. There are also cloud resources which give compute and storage capacity to anyone at low or no cost. This is a trend across many disciplines and even in the private sector.

lines 99-102: This point is misleading. You also are downloading large volumes of data, you are just automating it for the user. They would still need to wait while it downloads or they would benefit from you having pre-downloaded it for them. This is how the community does analysis already. Users can download data once into a project group directory on an institutional cluster where many scientist can perform their analysis. In the case of the CMIP6 archive, much of it has been copied onto NERSC drives where it is directly available to the community via a Jupyter notebook interface. There was even a multi-institutional hackathon [3] to collectively work to push results out faster. The point is that there are many ways around the need of downloading large amounts of data. If access to these institutional clusters is an issue, this is a need that the community should address.

line 118: Jupyter notebooks [4] are another widely used solution which you should reference.

lines 141ff: It is interesting that CAFE can deal with data sitting in different locations. However, I wonder how scalable this idea is. If the required data is large, then the runtime of TraceME will be dominated by download times. This may be acceptable for a relatively small analysis (few variables for a few models at monthly resolution), but could be on the order of days/months if higher temporal frequency is to be analyzed.

lines 152ff: Where is the web-based UI? It feels strange to see you advertise this 'shareable' technology and then not have access to explore just what it is. What are the limitations of what I can trace? Does it depend on what you have previously downloaded? Can I upload my own model output? Can I edit the analysis script that is run? Or does it rather run on the limited models you have predownloaded and only the analysis you have setup? If this is the case, a web UI seems superfluous. You could simply upload all possible results to a website for community perusal. In fact, this is what ILAMB does and how the service is most used.

lines 234-235: So the data must be moved to the central node, doesn't this mean download times will dominate your analysis? How is this computationally efficient?

lines 244-245: Does this mean that we are restricted to using models as they were uploaded a year ago? A lot of model data has been uploaded and updated. Or will the web execution of TraceME automatically query a search of ESGF and redownload these model outputs?

line 360: You should expand on what you mean by each of these terms.

* What does 'traceable' mean beyond the execution of your analysis? Why is this aspect of model evaluation so critical? * Does 'automatic' mean executable from a web form? If so, ILAMB has had this for 2 years in the work done by Mark Piper [5]. Also, on each commit to the master branch, ILAMB deploys automatically on Azure-pipelines [6], downloads observational data, and runs a test. Also as you mention, both ILAMB and ESMValTool have workflow that make the (parallel) computation of a huge suite of model evaluations automatic. If this is not what you consider 'automatic', then what is 'automatic' and why is what the community is doing insufficient? * What does 'shareable' mean? The ILAMB package generates a hierarchy of evaluation results that are browseable in a web page that you can distribute to the world by simply uploading it to a web-accessible location. If that is not 'shareable', what is and why is it so important to model evaluation? Furthermore, not every group wants a shareable solution, say for quick verification tests they do not want accessible.

lines 385f: I disagree with you that model evaluation needs to be more efficient. ILAMB may take a long time in serial execution, but this is why it was written to launch in

parallel on several institutional clusters or even a laptop/workstation. I am aware that the entire ILAMB CMIP5v6 comparison runs in a few hours. Given the decadal span between MIPs, I contend that the speed of our analysis is not the bottleneck. Beyond this, there are scripting tools and packages specifically designed to handle parallel and fast evaluation (see dask [7] and xarray [8] among others).

line 387: You argue that there is an 'absence' of automation and then explain how ILAMB and ESMValTool both implement it?

line 396: Unfortunately there is no substitution for technical training. You can setup a system like TraceME which automatically runs analysis. Yet someone has to setup and maintain that system. As software stacks change, it will break. Models will need to be added and updated. The analysis script will need to change. Others will want to upload their own scripts. How will they do this? There is a great amount of technical work that is needed to keep such a setup running and useful. What you have done is made running a relatively narrow task simple, which is by far the easiest part of the work.

lines 404ff: You have not solved the issue of data transfer, you have hidden it. And it is not really hidden either. When the user clicks on your web interface and then has to wait, perhaps days, while the data is downloaded to your central node, it will not feel terribly automatic.

[1] Koven, Hugelius, Lawrence, Wieder, Higher climatological temperature sensitivity of soil carbon in cold than warm climates. Nature Climate Change, October 2017, doi: 10.1038/NCLIMATE3421 [2] https://docs.nersc.gov/connect/jupyter/ [3] https://eos.org/science-updates/hackathon-speeds-progress-toward-climate-model-collaboration [4] https://jupyter.org/ [5] https://permamodel.github.io/pbs/ [6] https://azure.microsoft.com/en-us/services/devops/pipelines/ [7] https://dask.org/ [8] http://xarray.pydata.org/en/stable/

---

## Referee Comment (RC2) · Anonymous Referee #2 · 26 Jun 2020

In this work, the authors developed the TraceME system, in order to address what they argue are the three core challenges of ESM evaluation: the untraceable of model outputs, the lack of automatic algorithms and the high computational cost. They therefore built a cloud-based evaluation system, which, according to the authors, is traceable, automatic and sharable. The system was built on a previously established collaborative analysis framework of CAFE. I do believe that the traceability framework, which has been continuously developed by a few authors in this study since 2012, is a very useful one to expose model structure differences and errors in simulating land carbon cycle processes. But I am not convinced that substantial advances in terms of scientific model development have been made in this specific work to warrant its publication in

Geoscientific Model Development.

There is large room for improvement toward being more rigorous in writing and better logical flow in the present work. Very often, the authors either laid a too much wide background and then end up with a much narrower implementation, or used a lot vague expressions to justify the added value of their work. Throughout the whole text, a better and more rigorous justification for the novelty and usefulness of TraceME is needed, especially in a sense to the wider modeling community in contrast to those who are interested in traceability framework. Below are some major comments that lead me the above conclusions:

Major Comment #1: Line 23: 'the untraceable model outputs' pre-assumes the readers' knowledge on traceability framework and assumes traceability is foremost important in evaluating ESMs. I am not convinced on this. I believe every modeling group, when looking at their model performance in development cycles, would try to 'trace' the error into its underlying processes and understand the causes. In this sense, there is no model output that is 'untraceable'. The justification for the necessity of TraceME for the wider modeling community, and its usefulness in day-to-day model development has not been demonstrated in the paper.

One core argument for 'automatic' and 'sharable' evaluation platform would be to help identify model errors and improvement directions. If this is only for some key MIPs like CMIP5 or CMIP6, then it seems that analyzing the output on this platform by the authors and making the webpage available for different modeling groups would be sufficient. This would further raise doubts on whether there is value for this work to be published and for the tool to be available for the whole modeling community. There is a lack of evidence in the paper that modeling groups would indeed be interested to visit the platform and use it in their work. In the contrary, the figures contained inside make it more like a normal science paper. If by reading the paper figures, modelers would already have the information needed, I doubt they would visit the platform. Then the 'sharable' key feature would be not that useful either.

[Figure]

Major comment #2: The authors discussed in several places of the Introduction section the mounting challenges of evaluation of ESMs and cited the large volume of data from CMIP projects but ultimately nailed down only to its land component, or more specifically, the land carbon cycle component. In this case, the advantage of traceability seems only valid in evaluation of the land carbon cycle models. This point weakens the importance of their work and leaves the introduction scope of evaluation of ESMs (especially the 1st paragraph there) unmatched to what the authors actually delivered finally. Even for evaluating land carbon cycle models, I think the traceability framework oversimplifies the complexity of the land carbon cycle process. Disturbances, land use change and land management become increasingly important in carbon cycle models, can the traceability framework accommodate the differences in these factor among models? The conclusion in lines 77-78 seem unfair for other evaluation tools because the traceability framework is based completely on the idea of pool size and residence time, and finds its best application in carbon cycle models but not in others. The ESMs evaluation also includes those on hydrology, radiation and land-atmosphere interactions. The authors seemed ignoring these in their traceability framework.

Major comment #3: I downloaded the code provided at the end of the paper. There seems only a few python and R scripts with several hundred lines. There are not any user guides or documentation. No weblink for TraceME was provided in the paper either (I hope I did not miss it though). The modeling community is left only reading the paper and wonder how they can use this tool. This is at odds with what the authors claim that TraceME is 'sharable'.

Major comment #4: For a paper focusing on model development, descriptions on the technical aspects of the development, e.g., on the technical roadmap selection, implementation details, code structure and platform architecture, description of the key but new processes in contrast to previous model versions, usually take an important part in the paper. But the technical description on the TraceME development is rather weak in this paper. The only section on this topic might be Section 2.1. But the description

is vague and general. It is unclear what is the novelty in TraceME compared to CAFE, and which part of work has been done by engineering support and which by the authors, and what is the technological novelty. I cannot believe with the several hundred lines of python and R scripts provided by the authors in the 'Code Availability' section would make such a complex platform as described in the paper.

Major comment #5: Key arguments for TraceME by authors include automatic algorithms, sharable and saving the need to download data. The concept of 'automatic' is vague. For the results presented in the paper, I agree the authors make these figures automatically because the scripts must be extensively tested. But the authors do not show that beside what they have presented, if modeling groups want to use the platform practically, how much flexible and automatic could it be? If indeed it's useful, the data uploading and downloading would be unavoidable.

Minor comments:

There are many places the writing is causal and ambiguous and needs to be improved, to name a few examples:

Line 45-47: some articulations are needed here. Current statements are a little too general. Does 'their' in 46 refer to 'metrics', how can these metrics have 'indirect effects'? What are these 'indirect effects'?

Line 47-48: 'it is not independence among models' => unclear.

Line 49: '80% of the variance' => the variance of what ?

Line 55: dramatically => dramatic

Line 74: land information system => unclear what does this mean.

Line 109: it needs a new platform => a new platform is needed . . .

Line 113: automatic and shareable platform => "an" automatic and shareable platform

Line 189: the externally forces => external forcings ?

Line 190: is always deviate from = > please check the grammar here.

Line 251: that had been submitted results => 'been' should be removed.

Line 685: positive above the soil lines => 'soil' should be 'solid'

Line 594: composed into => decomposed into?

Line 360: needs to some new characteristics => check grammar

Line401-403: I don't see how the citation of Song 2019 fit here. Song et al. is based on site level which is at a completely different scale of what has been presented in the paper.

Line 104-105: the citation of data volume for CMIP5 and CMIP6 has not direct relevance. I guess nobody would download and analyze all the data for all variables. Focusing on several variables would not lead to download more data in CMIP6 than CMIP5 unless spatial resolution dramatically increases.

Line385-388: I understand 'computational efficiency' as how many tasks are done given a unit of computation resource. The author argued that automated computation increase efficiency, but this was not proved in the paper.

---

## Author Comment (AC1) · 8 Aug 2020

**Responses to Comments from Reviewer #1**

**General comments:**

*This paper discusses the so-called traceability analysis that the authors have published and applied since 2013. The motivation for the paper is the development of a cloud-based software built on the CAFE framework which implements their method and is launched via web interface. The authors position their analysis as the answer to what they consider are the deficiencies in the current state of model evaluation, describe how their software and algorithms work, and then show analysis results from a selection of CMIP6 models.*

Thank you very much for your careful reading and valuable suggestions.

**Comment 1:** *The first is that the novel portion of the paper, the TraceME software, is not a substantial advance in modeling science. TraceME is a web interface to an analysis script which runs on their server. There is some sophistication in that source data can exist on multiple nodes, but this is the CAFE framework and not TraceME itself. The authors neither provide a link to test out the operability of the software they describe or even a screenshot of the user interface. The reader is left to trust the authors that this software exists and is functional as they say it is.*

**Response:** Thank you for raising this important issue. TraceME has two significances: performance and traceability. Performance means the model evaluation process could be facilitated by just accessing to the online system, submitting analysis tasks, and getting analysis results back. Researchers do not need to download any original model data. Traceability means the model evaluation could be done based on the Traceability analysis capability.

Although CAFE framework does provide an online process environment where multiple model data servers can automatically collaborate with each other to fulfill users' analysis tasks, there are three specific enhancements that are just enabled in TraceME:

1. Multi-variable interaction processing to satisfy systematic traceability analysis. It consists of the task submission module (merging multi-variable into one task), the multi-variable preprocessing module, the data query module (multi-result systematic query), and the result maintenance module.

2. Providing a traceability analysis module and a systematic evaluation module based on python and R languages, and having them integrated with the CAFE framework.

3. Fine-tuning of the connection to the Python language in CAFE.

The TraceME (v1.0) system was deployed in a local intranet environment for internal usage only. It is now ready for public access. The website is http://traceme.org.cn. The screenshots below show the "data Search" page, task list page, and the analysis results page. Details about the workflow of TraceME are provided in the supplementary material (end of this text), which is expected to be incorporated into the manuscript during the revision.

[Figure]

Figure. R1. The screenshot of the Search page.

[Figure]

Figure. R2. The screenshot of the My Tasks page.

[Figure]

Figure R3. The screenshot of the results page.

**Comment 2:** *My second reservation is that a significant portion of the paper puts forth a viewpoint of the state and needs of model evaluation that is poorly supported. The authors use ambiguous terms such as 'traceable', 'shareable', 'indirect effects','high computational cost', and 'automatic'. They use these terms to describe their viewpoint on the deficiencies of current model evaluation, but do not say in detail what these terms mean.*

**Response:** Thanks for providing this comment. Since this question is highly related to your comment 4 and 12 below, we provided explanations for 'indirect effects' in the response to comment 4, and 'traceable', 'shareable', and 'automatic' in the response to comment 12.

We added more explanations on these terms during the revision in the *Introduction* and *Discussion* sections.

The term 'high computational cost' is not accurate. We have removed it during the revision.

**Specific Comments:**
**Comment 3:** *lines 22-24: You assert that the main challenge of using observations to evaluate ESMs is the untraceability of model outputs. It is not clear to me what this means precisely or why it is true. Why is it the 'main' challenge among many others?*

**Responses:** Thanks for raising this issue. The model evaluation process usually consists of three steps: downloading the model output data and archiving them locally, pre-processing the data to be suitable to be analyzed, utilizing a dedicated program to finish the evaluation.

Among many challenges in these three aspects that researchers need to address to fulfill model evaluation, data volume and traceability are major issues that we would like to highlight in this study. Taking the Coupled Model Intercomparison Project (CMIP) as an example, the total volume of CMIP Phase 5 (CMIP5) data is about 1.5PB, while the ongoing CMIP Phase 6 (CMIP6) data is expected to be 40-60 PB. Such a rapid explosion in data volume largely resulted from the increase in model complexity. As one featured component in CMIP models, land carbon cycle has been becoming highly complex during the past two decades. However, it's modeling uncertainty has been very high in CMIP5, which will still exist in the CMIP6 (Please see the below figure). Many recent studies have recognized that it is urgent and important to understand why these models perform so differently among the CMIP models (e.g., Bonan, et al., 2019). Indeed, a tool which can trace the model-to-model difference based on the big CMIP data is urgently needed. From the model evaluation perspective, it is highly needed to quantify the structural sources of the uncertainty within the complex models.

We agree with the reviewer that *Lines 22-24* are not clear. We have revised them to make the above point clearer.

[Figure]

Figure R4. The uncertainties of land carbon sink simulated by models from IPCC reports.

**Comment 4:** *lines 44-45: What does 'generally treat all metrics equally' mean? The referenced ILAMB package, for example, does not treat every statistical measure equally. Neither does it treat measures from all data sources equally. Also, what do you mean by 'indirect effects'? ILAMB also considers variable-to-variable relationships including metrics such as Koven's inferred carbon turnover time [1]. I am having trouble envisioning what the authors mean by this statement.*

**Responses:** Thanks for pointing out this issue. We have removed that statement in the revised version. To the best of our knowledge, only the ILAMB has used a weighted approach to score different models based on the data. The ILAMB is the current paradigmatic package leading the international model evaluation activities, so we introduced it as the novel effort rather than a traditional method in the second paragraph of the Introduction section. It is great that the Koven's carbon turnover time has been used as one scoring matric in the ILAMB. We have made it clear in this revised version.

**Comment 5:** *lines 56-59: It does not follow that 'an automated computation and shareable platform' is essential because of a increase in the amount of data. Environmental computing has utilized computing centers for many decades in response to computational and data costs. Furthermore, these computing centers are becoming more user-friendly. For example, NERSC now supports Jupyter notebooks [2] which allow you to script analysis on your browser without needing to move data around. There are also cloud resources which give compute and storage capacity to anyone at low or no cost. This is a trend across many disciplines and even in the private sector.*

**Responses:** We appreciate the reviewer for introducing the computing centers and related progress. We agree with the reviewer that data centers are becoming more user-friendly. In the revised version, we have removed the emphases on "automated and shareable" as the key features in the TraceME. The major novel characteristic of TraceME is traceability, and we have made it clearer in the new version. The examples of NERSC and cloud-based resources have been cited as one example in our revised manuscript.

***Comment 6:*** *lines 99-102: This point is misleading. You also are downloading large volumes of data, you are just automating it for the user. They would still need to wait while it downloads or they would benefit from you having pre-downloaded it for them. This is how the community does analysis already. Users can download data once into a project group directory on an institutional cluster where many scientist can perform their analysis. In the case of the CMIP6 archive, much of it has been copied onto NERSC drives where it is directly available to the community via a Jupyter notebook interface. There was even a multi-institutional hackathon [3] to collectively work to push results out faster. The point is that there are many ways around the need of downloading large amounts of data. If access to these institutional clusters is an issue, this is a need that the community should address.*

**Responses:** To avoid possible misleading, these lines have been removed during the revision. We greatly appreciate the reviewer for rising the new important issue on accessing to multiple institutional clusters of CMIP data. We have added a few sentences in the *Discussion* section to discuss this issue. In brief, we agree with the reviewer that institutional clusters are continually enriched. Taking CMIP6 for example, there are 30 data nodes available right now (https://esgf-node.llnl.gov/search/cmip6/). According to the WGCM Infrastructure Panel (https://www.wcrp-climate.org/wgcm-cmip/wip), no single institution has committed 40-60PB storage to host all the CMIP6 data in one place. As highlighted by the reviewer, federating multiple data nodes/ institutional clusters to enable a new collaborative analysis environment is needed. We also have discussed how the TraceME system can bring the traceability analysis to

benefit the CMIP6 evaluations across different institutional clusters.

**Comment 7:** *line 118: Jupyter notebooks [4] are another widely used solution which you should reference.*

**Responses:** Thanks for the suggestion. Jupyter notebook has been referenced in the revised version.

**Comment 8:** *lines 141ff: It is interesting that CAFE can deal with data sitting in different locations. However, I wonder how scalable this idea is. If the required data is large, then the runtime of TraceME will be dominated by download times. This may be acceptable for a relatively small analysis (few variables for a few models at monthly resolution), but could be on the order of days/months if higher temporal frequency is to be analyzed.*

**Responses:** Thank you for raising this issue. TraceME follows the design principles of CAFE: 1) analysis functions are available on the data server and are accessible through a website; 2) the analyses are done remotely on the data server and users do not need to download the data and archive them locally; 3) users only need to download the analytical results as pictures, graphs, or text files. The files in native formats, e.g. NetCDF files, are also available upon request; 4) multiple data nodes could collaborate with each other so that data archived on different nodes could be analyzed automatically and collaboratively if they are specified in one request. The download time is dramatically minimized.

The reason that we selected the CAFE framework is that it is capable of linking multiple data nodes together to fulfill users' data analysis requests. For example, there are 30 data nodes in CMIP6. Except for some supernodes that could replicate some selected data archived in other nodes, data hosted in one node won't be available in other nodes. Since researchers always need to intercompare multiple models, they need to download CMIP6 data of interest from individual data nodes individually.

CAFE enables a new way to access to CMIP6 data. It is capable of automatically linking these nodes together to build a federation. It provides a data search panel for researchers to search for CMIP6 data of interest, presents a list of built-in data analysis functions to select. Researchers only need to submit a model intercomparison request. CAFE can automatically locate the data nodes where the CMIP6 data need to be analyzed reside, forward the request to those data nodes. Each data node can then finish the request automatically, and then forward the result back to the node where the request was forwarded. All the results will be finally presented to researchers in the end. During this whole process, researchers neither download the original data nor know where the data actually reside. More details about the data downloading can be found in Xu et al. (2019).

One further novel development in TraceME is that the CMIP6 model data has been pre-processed to be "yearly" data, so that the volume of data needs to be analyzed in TraceME has been dramatically reduced. We have made it clearer in the *Discussion* section of the revised version.

**Comment 9:** *lines 152ff: Where is the web-based UI? It feels strange to see you advertise this 'shareable' technology and then not have access to explore just what it is. What are the limitations of what I can trace? Does it depend on what you have previously downloaded? Can I upload my own model output? Can I edit the analysis script that is run? Or does it rather run on the limited models you have predownloaded and only the analysis you have setup? If this is the case, a web UI seems superfluous. You could simply upload all possible results to a website for community perusal. In fact, this is what ILAMB does and how the service is most used.*

**Responses:** We are truly sorry for the inconvenience that this has caused. We now have made the TraceME available at this URL: http://traceme.org.cn. Details about this website have provided in the response to your first comment, and the workflow of TraceME has been described in the supplementary material (end of this text).

To facilitate the traceability analysis for the CMIP6 data in TraceME, we have pre-downloaded land model variables for all the CMIP6 models. Yes, the data that can be analyzed are those available in the TraceME system. At its current form, TraceME just pre-downloaded the selected CMIP6 model data to enable this new traceability analysis capability for the community. We are building a large CAFE federation in China by federating two CMIP6 data nodes and setting two more ones. Once it is ready, the TraceME system will be able to provide analysis capability for all the BCC and CIESM land process model output. Although the concept of CAFE is general-purpose, it focuses on the CMIP5 and CMIP6 data at its current stage. Therefore, it relies on the file naming and organization conventions that CMIP5 and CMIP6 follows. Uploading users' own model output cannot be supported at the moment, as the main focus of CAFE is enabling easy access to data servers. As the internal data management module can only recognize CMIP5 or CMIP6 datasets, uploading CMIP5 or CMIP6 compliant datasets is doable.

We really appreciate the reviewer for sharing with us the experience of ILAMB. We have recently learned the ILAMB package and discovered many novels and great features that can facilitate the application of TraceME.

**Comment 10:** *lines 234-235: So the data must be moved to the central node, doesn't this mean download times will dominate your analysis? How is this computationally efficient?*

**Responses:** Sorry for the confusion. There is no need to transfer data from data nodes to the central node. In this version, we have removed the statements about 'computationally efficient', and provide more explanations on the role of the central node during the revision.

**Comment 11:** *lines 244-245: Does this mean that we are restricted to using models as they were uploaded a year ago? A lot of model data has been uploaded and updated. Or will the web execution of TraceME automatically query a search of ESGF and redownload these model outputs?*

**Responses:** Thank you for raising this important issue. In the current version of TraceME (v1.0), we pre-download CMIP6 model data and manually keep the modeling data updated based on the ESGF server. We are currently adding more CMIP models to the TraceME website. In the future version, we are actually working on two data downloading functions. One is building a large CAFE federation in China by federating two CMIP6 data nodes and setting two more portals. Once it is ready, the TraceME system will be able to provide analysis capability for all the BCC and CIESM model data without pre-downloading any of them. The other is having a close collaboration with WGGM Infrastructure Panel and ESGF to enable a new collaborative analysis environment for all the CMIP6 nodes. If this is the case, CMIP6 data inter-comparison work would be dramatically facilitated, and the need to download a large volume of original CMIP6 data would be maximally minimized. Further introduction to the workflow of TraceME is ready in supplementary materials (end of this text), which is expected to be incorporated into the manuscript during the revision.

**Comment 12:** *line 360: You should expand on what you mean by each of these terms.*

*\* What does 'traceable' mean beyond the execution of your analysis? Why is this aspect of model evaluation so critical?*

*\* Does 'automatic' mean executable from a web form? If so, ILAMB has had this for 2 years in the work done by Mark Piper [5]. Also, on each commit to the master branch, ILAMB deploys automatically on Azurepipelines [6], downloads observational data, and runs a test. Also as you mention, both ILAMB and ESMValTool have workflow that make the (parallel) computation of a huge suite of model evaluations automatic. If this is not what you consider 'automatic', then what is 'automatic' and why is what the community is doing insufficient?*

*\* What does 'shareable' mean? The ILAMB package generates a hierarchy of evaluation results that are browseable in a web page that you can distribute to the world by simply uploading it to a web-accessible location. If that is not 'shareable', what is and why is it so important to model evaluation? Furthermore, not every group wants a shareable solution, say for quick verification tests they do not want accessible.*

**Responses:** Thanks for the suggestions. We have made substantial modifications to this part according to the great suggestions.

First, in our study, 'traceable' refers to the traceability analysis method, which can trace the structural sources of the uncertainties of key model components in land models. For example, carbon storage dynamics can be decomposed into carbon storage capacity and potential, and NPP can be decomposed into GPP and CUE. This method can systematically quantify the structural sources of uncertainty in land models. The increasingly complex model and the requirement for model development also drive model evaluation to provide more instructive information to better understand the sources of uncertainty. Traceable methods would be a great development and supplement to the traditional methods of statistical comparison. For example, the results of our traceability analysis can be used as a supplement to the scoring criteria of ILAMB. Besides, for example, ESMValTool (v1.0) uses a log file to record all information on a task, which is also considered as the traceability of the results (Eyring et al., 2016), while TraceME (v1.0) adopts the database (MySQL) to record it, which belongs to the content of technical description and will not be mentioned in this discussion.

Second, 'automatic' in our study refers to the web-based workflow that can automatically run the processes, such as searching, downloading, managing, preprocessing, and analyzing data. As the reviewer has mentioned, many other model evaluation tools or web-based systems, such as ILAMB and ESMValTool, also can run automatically execute a workflow for model evaluations. 'automatic' is not a novel property for model evaluation and just a technical function of TraceME. Thus, we plan to remove 'automatic' in the revision.

Third, 'shareable' in our study mainly means that model evaluation requires a platform for sharing data, which can reduce the time for users spending on repeating searching, downloading, managing, and preprocessing data, to improve the efficiency of model evaluation community. In the revised version, we have removed the emphasis on the 'shareable' feature, and only discussed the benefits of the TraceME platform for the evaluation community. We appreciate the reviewer for the introduction about the characteristics of ILAMB and have added it in the *Discussion* section of the revised manuscript.

**Comment 13:** *lines 385f: I disagree with you that model evaluation needs to be more efficient. ILAMB may take a long time in serial execution, but this is why it was written to launch in parallel on several institutional clusters or even a laptop/workstation. I am aware that the entire ILAMB CMIP5v6 comparison runs in a few hours. Given the decadal span between MIPs, I contend that the speed of our analysis is not the bottleneck. Beyond this, there are scripting tools and packages specifically designed to handle parallel and fast evaluation (see dask [7] and xarray [8] among others).*

**Responses:** We agree with you on this comment, especially what the ILAMB team has

promoted in recent years. References to these packages have been added. We also removed this statement.

**Comment 14:** *line 387: You argue that there is an 'absence' of automation and then explain how ILAMB and ESMValTool both implement it?*

**Response:** We have removed the statements on automation in the revised version.

**Comment 15:** *line 396: Unfortunately there is no substitution for technical training. You can setup a system like TraceME which automatically runs analysis. Yet someone has to setup and maintain that system. As software stacks change, it will break. Models will need to be added and updated. The analysis script will need to change. Others will want to upload their own scripts. How will they do this? There is a great amount of technical work that is needed to keep such a setup running and useful. What you have done is made running a relatively narrow task simple, which is by far the easiest part of the work.*

**Responses:** Thanks for this very important comment. We have removed the statements on the aspects of automatic and sharable in this version, so these sentences have been removed in this version. We agree with the reviewer that there many challenges to keep model evaluation systems like TraceME. The great package of ILABM has provided us many ideas to improve the TraceME. Also, one real challenge approaching to us is that how researchers worldwide could effectively finish their CMIP6 data analysis work when the total volume of CMIP6 data could be 40-60PB, while it is just about 1.5 PB for CMIP5. We also agree with the reviewer that what TraceME has pre-downloaded is just a very small part of this huge data archive. We have setup a dedicated CMIP6 data download speed test website (http://www.cmip6speedtest.cn/) to know how fast downloading CMIP6 data in a different part of the world could be. Browser-based data transfer speed testing against 28 data nodes from 44 testing cities has been finished. The mean download speed is just 6.12 MB/s.

One basic idea is that CMIP6 data nodes are expected to take more responsibility other than just disseminating the model data. If researchers have to stick with the original scenario that they need to download, archive and process the CMIP6 model data by themselves, then finding a win-win solution is impossible. Dealing with this issue is a long-term goal of the CAFE system, and the TraceME will gradually adopt some applicable features in its future versions.

**Comment 16:** *lines 404ff: You have not solved the issue of data transfer, you have hidden it. And it is not really hidden either. When the user clicks on your web interface and then has to wait, perhaps days, while the data is downloaded to your central node, it will not feel terribly automatic.*

**Responses:** We are sorry for not making this part unclear to you. TraceME does pre-downloaded several important variables-related CMIP6 land model data. Therefore, during the traceability analysis process, there is no need to further download CMIP6 data. The workflow of TraceME is provided in supplementary materials (end of this text). This part is going to be appended to the manuscript during the revision. We have also seriously discussed the ideas from the reviewer's suggestions based on ILAMB. We have realized the advantages of ILAMB for evaluating CMIP models, and we will try to collaborate with the ILAMB team for facilitating the model evaluations on Earth system models.

**References**

Collier et al. The International Land Model Benchmarking (ILAMB) System: Design, Theory, and Implementation, J. Adv. Model. Earth. Sy., 10, 2731-2754, 2018.

Eyring et al. ESMValTool (v1.0) – a community diagnostic and performance metrics tool for routine evaluation of Earth system models in CMIP, Geosci. Model. Dev., 9, 1747-1802, 2016.

Schwalm et al. A model-data intercomparison of $CO_2$ exchange across North America: Results from the North American Carbon Program site synthesis, J. Geophys. Res., 115, 2010.

Xia et al. Traceable components of terrestrial carbon storage capacity in biogeochemical models, Global. Change. Biol., 19, 2104-2116, 2013.

Xu et al. A collaborative analysis framework for distributed gridded environmental data, Environ. Model. Softw., 111, 324-339, 2019.

**Supplementary materials: the scientific workflow of TraceME**

Within the workflow of TraceME, user can filter data of interest from the entire system, and the selected data is then packaged into a task and delivered to the assigned work node for data processing, which includes data pre-processing, traceability analysis, and evaluation, and finally, the evaluation results are output and visualized for the users (Fig. S1). The scientific workflow is essential for TraceME to realize online automated model evaluation. The detail of the workflow will be described below.

[Figure]

Figure. S1 The workflow of TraceME.

The function of TraceME providing for users to filter data mainly comes from the collaborative framework of CAFE and its various web application-programming interfaces (API). This function includes data source collection, data query and filtering, and submitting the task of selected data. Data is stored on individual work nodes, which can automatically parse data information to the database of work node according to the root directory of the data source by a specific API (http://{host}: {port}/{work node name}/web/parser). Then the central node collects all data information from all work nodes and provides it to the "Search" page for

users to query and filter data by APIs (Fig. S1). After users submit their selected data, the system packages all the information of data into a task for subsequent processing. TraceME focuses on evaluating models systematically with multiple variables, while the framework of CAFE is that one variable is a task. Thus, we have added new modules to support the task with multiple variables and multiple models.

Data processing in TraceME mainly includes three steps: data preprocessing, traceability analysis, and evaluation (Fig. S1). Among them, data preprocessing is mainly inherited from the CAFE framework, and we modify it to accommodate the multi-variable processing and archiving. When a task is submitted, the central node will arrange a work node for data processing. For the system to process all kinds of data uniformly, the data needs to be preprocessed first, which includes time and space extraction based on user selection and spatial resolution conversion (1x1°) by calling the tools of NetCDF Operators (NCO) and Climate Data Operator (CDO). In this version of TraceME, according to the needs of systematic traceability model evaluation, the variables from models include key process variables (NPP, GPP, and carbon storage) and forcing factors (temperature and precipitation). These preprocessed data are then submitted to the traceability analysis module written by python. With the framework of traceability analysis, land carbon storage can be decomposed into various traceable components, such as carbon storage capacity, carbon storage potential, residence time, carbon use efficiency (CUE), and baseline residence time along a temporal or spatial axis. These components are the primary objects for evaluation, and in the current version, the model evaluation includes the standard deviation of these components among models and the variance contribution of these components to the uncertainty of land carbon storage based on a hierarchical partitioning method that is written by R language.

After traceability analysis and evaluation, TraceME (v1.0) provides systematic results about the evaluation, including the figures and nc-format files of each traceable components and their variance contribution to the uncertainty of simulated land carbon storage by the models (Fig. S1). This involves task management, structured results storage, and visualization. Each task in TraceME (v1.0) has a unique task ID and is recoded the ownership of the task, the

information about data and work node, the status of processing, and the results through the database (MySQL). The "My tasks" page of TraceME displays the status of the task and the structured results, and it also provides the available links to download them for users via various API (Fig. S1).

---

## Author Comment (AC2) · 8 Aug 2020

**Responses to Comments from Reviewer #2**

**General comments:**

*In this work, the authors developed the TraceME system, in order to address what they argue are the three core challenges of ESM evaluation: the untraceable of model outputs, the lack of automatic algorithms and the high computational cost. They therefore built a cloud-based evaluation system, which, according to the authors, is traceable, automatic and sharable. The system was built on a previously established collaborative analysis framework of CAFE. I do believe that the traceability framework, which has been continuously developed by a few authors in this study since 2012, is a very useful one to expose model structure differences and errors in simulating land carbon cycle processes. But I am not convinced that substantial advances in terms of scientific model development have been made in this specific work to warrant its publication in Geoscientific Model Development. There is large room for improvement toward being more rigorous in writing and better logical flow in the present work. Very often, the authors either laid a too much wide background and then end up with a much narrower implementation, or used a lot vague expressions to justify the added value of their work. Throughout the whole text, a better and more rigorous justification for the novelty and usefulness of TraceME is needed, especially in a sense to the wider modeling community in contrast to those who are interested in traceability framework.*

**Response:** We highly appreciate the critical comments on this work. After carefully studying the comments from the reviewer, we have substantially revised the manuscript. We have tried our best to revise the manuscript to show the novelty and scientific value of the TraceME platform. We hope the revised manuscript is not only useful for people who are interested in the traceability framework but also helpful for a wider modeling community. Please find our point-by-point replies below.

*Below are some major comments that lead me the above conclusions:*

**Major Comment:**
**Comment 1:** *Line 23: 'the untraceable model outputs' pre-assumes the readers' knowledge on traceability framework and assumes traceability is foremost important in evaluating ESMs. I am not convinced on this. I believe every modeling group, when looking at their model performance in development cycles, would try to 'trace' the error into its underlying processes and understand the causes. In this sense, there is no model output that is 'untraceable'. The justification for the necessity of TraceME for the wider modeling community, and its usefulness in day-to-day model development has not been demonstrated in the paper.*

**Response:** Thanks for pointing this issue out. We have made a substantial revision to make the traceability clearer before the introduction of TraceME. First, a few sentences have been added in the introduction to define the method of traceability analysis and the reasons for developing it into a platform. Then, a description of the scientific workflow in TraceME is provided in the supplementary materials (end of this text), and more technical descriptions about TraceME have been incorporated. Furthermore, we have demonstrated the necessity and usefulness of TraceME for the modeling community. We showed that TraceME is useful not only for MIPs but also for specific modeling groups. For example, the TraceME has been applied to CLIM5.0 to evaluate the effects of different climate forcings (i.e., CRUNCEP and GSWP) on the simulated land C storage dynamics. As shown by the following figure, there is a 2-fold difference in global C storage capacity in CLM5.0 between the forcings of CRUNCEP and

GSWP. Such difference is jointly contributed from net primary productivity and C residence time.

[Figure]

Figure R1. The results of CLM5.0 with two different forcings (CRUNCEP and GSWP) come from TraceME. (a) Land carbon dynamics, carbon storage is decomposed into carbon storage capacity and potential. (b) Carbon storage capacity is decomposed into NPP and residence time. (c) NPP is decomposed into CUE and GPP. (d) Residence time is decomposed into environmental scalars and baseline residence time. All simulated data is the normal simulation from 1921 to 1940 after the spin-up.

**Comment 2:** *One core argument for 'automatic' and 'sharable' evaluation platform would be to help identify model errors and improvement directions. If this is only for some key MIPs like CMIP5 or CMIP6, then it seems that analyzing the output on this platform by the authors and making the webpage available for different modeling groups would be sufficient. This would further raise doubts on whether there is value for this work to be published and for the tool to be available for the whole modeling community. There is a lack of evidence in the paper that modeling groups would indeed be interested to visit the platform and use it in their work. In the contrary, the figures contained inside make it more like a normal science paper. If by reading the paper figures, modelers would already have the information needed, I doubt they would visit the platform. Then the 'sharable' key feature would be not that useful either.*

**Response:** Based on the suggestions from Reviewer #1, we have removed the highlights of "automatic" and "sharable" in the revised version. The main focus of the revised manuscript is

traceability analysis. We added one more case of CLM5.0 to show that the TraceME is applicable to a specific model to help the impact of different climate forcings. We believe the publication of the TraceME platform is helpful for the readers due to four reasons. First, during our collaboration with different CMIP model teams, we realized that many modelers know their models well on some specific processes, such as GPP or total carbon storage, but usually cannot explain why those processes are different from other models. Second, it still hard to understand how different versions of a specific model simulate different land carbon cycles from CMIP5 to CMIP6. Third, for many readers who use the CMIP results but not run the models, the TraceME platform is very useful for them to identify the key uncertainty components in different regions. Lastly, the large uncertainty issue if also emerged in other components in Earth system models, such as the hydrological cycle and nutrient cycle. The publication of the TraceME platform might be helpful for them to develop traceable tools to evaluate their models.

In a recent virtual training course organized by Northern Arizona University, we used the TraceME to teach the uncertainty analysis of CMIP models. Based on their feedback, we found this online version of the manuscript is very helpful to guide the trainees to understand the scientific importance of the model uncertainty and its traceability analysis.

**Comment 3:** *The authors discussed in several places of the Introduction section the mounting challenges of evaluation of ESMs and cited the large volume of data from CMIP projects but ultimately nailed down only to its land component, or more specifically, the land carbon cycle component. In this case, the advantage of traceability seems only valid in evaluation of the land carbon cycle models. This point weakens the importance of their work and leaves the introduction scope of evaluation of ESMs (especially the 1st paragraph there) unmatched to what the authors actually delivered finally. Even for evaluating land carbon cycle models, I think the traceability framework oversimplifies the complexity of the land carbon cycle process. Disturbances, land use change and land management become increasingly important in carbon cycle models, can the traceability framework accommodate the differences in these factors among models? The conclusion in lines 77-78 seem unfair for other evaluation tools because the traceability framework is based completely on the idea of pool size and residence time, and finds its best application in carbon cycle models but not in others. The ESMs evaluation also includes those on hydrology, radiation and land-atmosphere interactions. The authors seemed ignoring these in their traceability framework.*

**Response:** Thanks for the suggestions. We have removed the statement in lines 77-78 in the revised version. The reviewer has raised three important concerns in this comment, including why only focus on the land carbon cycle, how to consider disturbances and land use change in the traceability framework, and how to apply the traceability analysis to other components of ESMs. We do appreciate the reviewer for these important questions, which have been deeply discussed in the revised version. Below please find our brief replies:

First, the current version of TraceME focuses on the land carbon cycle mainly due to two reasons. One reason is the large uncertainty of the land carbon cycle in the recent CMIPs, and the other reason is that the traceability analysis has its theoretical foundation on the land carbon cycle. We are testing similar traceability analyses on other components of the ESMs, but some theoretical developments are still needed. In the revised version, we have revised the introduction to narrow the scope to the land carbon cycle. We emphasized that traceability is also an important need for evaluating other highly uncertain components in the current generation of ESMs.

Second, the theoretical basis of the traceability analysis could be traced back to Luo & Weng (2011), which has demonstrated that the land carbon cycle is a dynamic disequilibrium system. The dynamic disequilibrium of the land carbon cycle is jointly driven by the internal properties and external forcings of the ecosystem carbon cycle. The traceability analysis is mainly developed on the internal properties of the land carbon cycle, which can be described as a matrix equation. The external forcings, such as disturbances and land use change, can influence the different components in this equation. The following matrix equation describes the land carbon cycle in the CLM5.0, and it shows that the external forcings can affect carbon dynamics through different components or processes. The TraceME is developed based on such matrix equations, and it can be used to evaluate the impacts of disturbances and land use changes if the model provides the simulations with forcings with and without disturbances or land use changes. We have added one paragraph to discuss this topic in the revised version.

$$\frac{d}{dt}X(t) = (A_{ph}(t)K_{ph}(t) + A_{gm}(t)K_{gm}(t) + A_{fi}(t)K_{fi}(t))X(t) + B(t)F(t)$$

where, from left to right, the annotations read: C transfer of phenology / C turnover of phenology ($A_{ph}(t)K_{ph}(t)$), C transfer of gap mortality / C turnover of gap mortality ($A_{gm}(t)K_{gm}(t)$), C transfer of fire / C turnover of fire ($A_{fi}(t)K_{fi}(t)$), pool state ($X(t)$), allocation and input ($B(t)F(t)$).

Figure R2. The matrix equation for the carbon dynamics of CLM5.0.

Third, TraceME (v1.0) has not considered all terrestrial processes, such as hydrology, radiation, and land-atmosphere interactions. We are working hard to incorporate other processes such as hydrological and nutrient processes. We have added more discussion about the limitations of this version of TraceME and highlight other processes in future developments.

**Comment 4:** *I downloaded the code provided at the end of the paper. There seems only a few python and R scripts with several hundred lines. There are not any user guides or documentation. No weblink for TraceME was provided in the paper either (I hope I did not miss it though). The modeling community is left only reading the paper and wonder how they can use this tool. This is at odds with what the authors claim that TraceME is 'sharable'.*

**Response:** This is our mistake to ignore the 'Code Availability' section. We have ported the TraceME system from the local server (which is not easily accessible from the external network) to an external server so that reader can access it. The address of TraceME is http://traceme.org.cn. We also have provided the complete code on GitHub (https://github.com/ECNU-RCGCEF/TraceME).

**Comment 5:** *For a paper focusing on model development, descriptions on the technical aspects of the development, e.g., on the technical roadmap selection, implementation details, code structure and platform architecture, description of the key but new processes in contrast to previous model versions, usually take an important part in the paper. But the technical description on the TraceME development is rather weak in this paper. The only section on this topic might be Section 2.1. But the description is vague and general. It is unclear what is the novelty in TraceME compared to CAFE, and which part of work has been done by engineering support and which by the authors, and what is the technological novelty. I cannot believe with the several hundred lines of python and R scripts provided by the authors in the 'Code Availability' section would make such a complex platform as described in the paper.*

**Response:** Thanks for the suggestions. We have provided a more detailed technical description of TraceME in the revised version. The major revisions include the following aspects:

First, we introduced more details about CAFE infrastructure and the specific improvement in TraceME in *Section 2.1*. TraceME is developed based on CAFE. CAFE is a collaborative analysis environment where multiple data servers could work together to fulfill users' requests automatically. During the data analysis process, users only need to find data of interest, select analysis functionality to use, define analysis parameters, submit analysis tasks, and finally get the results. The logical structure of the CAFE system consists of one central node, several working nodes (data servers), and several web portals. The central node maintains descriptive information about all the data archived in each working node to support the data query. The working node is responsible for the analysis of the model data, according to users' requests. Web portals provide a browser-based GUI for researchers to interact with the whole CAFE federation.

Since it is hard to meet the requirement of systematic multivariate analysis in CAFE, we further develop the TraceME system to perform dedicated Traceability analysis for CMIP6 land models. The major enhancements that TraceME has enabled are the following:

1. Multi-variable interaction processing to satisfy systematic traceability analysis. It involves the task submission module (merging multi-variable into one task), adding a multi-variable preprocessing module, database module (multi-result systematic query), and structured results storage.
2. Deployment of the traceability analysis module and a systematic evaluation module based on python and R languages into CAFE
3. Fine-tuning of the connection to the Python language in CAFE.

Second, a description of the scientific workflow in TraceME is provided. We have incorporated it into the manuscript during the revision.

Third, we have setup a publicly accessible website of TraceME at this URL http://traceme.org.cn. Traceability analysis code is on the GitHub: https://github.com/ECNU-RCGCEF/TraceME.

**Comment 6:** *Key arguments for TraceME by authors include automatic algorithms, sharable and saving the need to download data. The concept of 'automatic' is vague. For the results presented in the paper, I agree the authors make these figures automatically because the scripts must be extensively tested. But the authors do not show that beside what they have presented, if modeling groups want to use the platform practically, how much flexible and automatic could it be? If indeed it's useful, the data uploading and downloading would be unavoidable.*

**Response:** Thanks for pointing this issue out. We have removed the discussions on 'automatic' and 'shareable' in the revision. Instead, we have added discussions on some new technical issues, such as how to federate multiple institutional clusters of CIMP data. These issues could be more important for developing model evaluation tools like TraceME. Then, a description of the scientific workflow in TraceME is provided in the supplementary materials (end of this text). We have incorporated it into the manuscript in the revision.

**Minor comments:**

**Comment 7:** *Line 45-47: some articulations are needed here. Current statements are a little too general. Does 'their' in 46 refer to 'metrics', how can these metrics have 'indirect effects'? What are these 'indirect effects'?*

**Response:** Thanks for pointing out this issue. We have revised the content in *Line 45-47* to:

"For example, the traditional methods used in model evaluation, are mainly using statistical approaches to measure the performance of models and generally treat all model components and their different metrics equally, but this ignores the relative contributions of them on model performance (Schwalm et al., 2010; Xia et al., 2013)."

**Comment 8:** *Line 47-48: 'it is not independence among models' => unclear.*

**Response:** We have removed it and revised 'it is not independence among models' to 'models share their components and are not independent of each other'.

**Comment 9:** *Line 49: '80% of the variance' => the variance of what ?*

**Response:** We have revised 'variance' to 'uncertainty' in the revised manuscript.

**Comment 10:** *Line 55: dramatically => dramatic*

**Response:** Done as suggested in the revised manuscript.

**Comment 11:** *Line 74: land information system => unclear what does this mean.*

**Response:** We have removed the 'land information system' in the revised manuscript.

**Comment 12:** *Line 109: it needs a new platform => a new platform is needed : : :*

**Response:** Done as suggested in the revised manuscript.

**Comment 13:** *Line 113: automatic and shareable platform => "an" automatic and shareable platform*

**Response:** Done as suggested in the revised manuscript.

**Comment 14:** *Line 189: the externally forces => external forcings ?*

**Response:** Done as suggested in the revised manuscript.

**Comment 15:** *Line 190: is always deviate from = > please check the grammar here.*

**Response:** We have revised 'so the $X_C$ is always deviate from' to 'the $X_C$ is always deviated from' in the revised manuscript.

**Comment 16:** *Line 251: that had been submitted results => 'been' should be removed.*

**Response:** Done as suggested in the revised manuscript.

**Comment 17:** *Line 685: positive above the soil lines => 'soil' should be 'solid'*

**Response:** Done as suggested in the revised manuscript.

**Comment 18:** *Line 594: composed into => decomposed into?*

**Response:** Is it 'composed into' in *Line 694*? We have revised 'composed into' in *Line 694* to 'decomposed into'.

**Comment 19:** *Line 360: needs to some new characteristics => check grammar*

**Response:** We have removed 'to'.

**Comment 20:** *Line401-403: I don't see how the citation of Song 2019 fit here. Song et al. is based on site level which is at a completely different scale of what has been presented in the*

*paper.*

**Response:** We have removed the citation of *Song et al. 2019* in *Line 401-403*.

**Comment 21:** *Line 104-105: the citation of data volume for CMIP5 and CMIP6 has not direct relevance. I guess nobody would download and analyze all the data for all variables. Focusing on several variables would not lead to download more data in CMIP6 than CMIP5 unless spatial resolution dramatically increases.*

**Response:** Thanks for the suggestion. We have removed the content in Line 104-105.

**Comment 22:** *Line385-388: I understand 'computational efficiency' as how many tasks are done given a unit of computation resource. The author argued that automated computation increase efficiency, but this was not proved in the paper.*

**Response:** We will remove the discussion about 'computational efficiency' in the revised manuscript.

**References**

Luo, Y., Weng, E.: Dynamic disequilibrium of the terrestrial carbon cycle under global change, Trends. Ecol. Evol., 26, 96-104, 2011.

Schwalm et al.: A model-data intercomparison of $CO_2$ exchange across North America: Results from the North American Carbon Program site synthesis, J. Geophys. Res., 115, 2010.

Xia et al.: Traceable components of terrestrial carbon storage capacity in biogeochemical models, Global. Change. Biol., 19, 2104-2116, 2013.

**Supplementary materials: the scientific workflow of TraceME**

Within the workflow of TraceME, user can filter data of interest from the entire system, and the selected data is then packaged into a task and delivered to the assigned work node for data processing, which includes data pre-processing, traceability analysis, and evaluation, and finally, the evaluation results are output and visualized for the users (Fig. S1). The scientific workflow is essential for TraceME to realize online automated model evaluation. The detail of the workflow will be described below.

[Figure]

Figure. S1 The workflow of TraceME.

The function of TraceME providing for users to filter data mainly comes from the collaborative framework of CAFE and its various web application-programming interfaces (API). This function includes data source collection, data query and filtering, and submitting the task of selected data. Data is stored on individual work nodes, which can automatically parse data information to the database of work node according to the root directory of the data source by a specific API (http://{host}: {port}/{work node name}/web/parser). Then the central node collects all data information from all work nodes and provides it to the "Search" page for

users to query and filter data by APIs (Fig. S1). After users submit their selected data, the system packages all the information of data into a task for subsequent processing. TraceME focuses on evaluating models systematically with multiple variables, while the framework of CAFE is that one variable is a task. Thus, we have added new modules to support the task with multiple variables and multiple models.

Data processing in TraceME mainly includes three steps: data preprocessing, traceability analysis, and evaluation (Fig. S1). Among them, data preprocessing is mainly inherited from the CAFE framework, and we modify it to accommodate the multi-variable processing and archiving. When a task is submitted, the central node will arrange a work node for data processing. For the system to process all kinds of data uniformly, the data needs to be preprocessed first, which includes time and space extraction based on user selection and spatial resolution conversion (1x1°) by calling the tools of NetCDF Operators (NCO) and Climate Data Operator (CDO). In this version of TraceME, according to the needs of systematic traceability model evaluation, the variables from models include key process variables (NPP, GPP, and carbon storage) and forcing factors (temperature and precipitation). These preprocessed data are then submitted to the traceability analysis module written by python. With the framework of traceability analysis, land carbon storage can be decomposed into various traceable components, such as carbon storage capacity, carbon storage potential, residence time, carbon use efficiency (CUE), and baseline residence time along a temporal or spatial axis. These components are the primary objects for evaluation, and in the current version, the model evaluation includes the standard deviation of these components among models and the variance contribution of these components to the uncertainty of land carbon storage based on a hierarchical partitioning method that is written by R language.

After traceability analysis and evaluation, TraceME (v1.0) provides systematic results about the evaluation, including the figures and nc-format files of each traceable components and their variance contribution to the uncertainty of simulated land carbon storage by the models (Fig. S1). This involves task management, structured results storage, and visualization. Each task in TraceME (v1.0) has a unique task ID and is recoded the ownership of the task, the

information about data and work node, the status of processing, and the results through the database (MySQL). The "My tasks" page of TraceME displays the status of the task and the structured results, and it also provides the available links to download them for users via various API (Fig. S1).